# Multi-scale structures of the mammalian radial spoke and divergence of axonemal complexes in ependymal cilia

Xueming Meng [1,7], Cong Xu[1,7], Jiawei Li[1,7], Benhua Qiu[2,7], Jiajun Luo[2,7], Qin Hong [1], Yujie Tong[1], Chuyu Fang[2], Yanyan Feng[3], Rui Ma[4], Xiangyi Shi[4], Cheng Lin[1], Chen Pan[5], Xueliang Zhu [2,6] ✉, Xiumin Yan [3] ✉ & Yao Cong [1,6] ✉

Radial spokes (RS) transmit mechanochemical signals between the central pair (CP) and axonemal dynein arms to coordinate ciliary motility. Atomic-resolution structures of metazoan RS and structures of axonemal complexes in ependymal cilia, whose rhythmic beating drives the circulation of cerebrospinal fluid, however, remain obscure. Here, we present near-atomic resolution cryo-EM structures of mouse RS head-neck complex in both monomer and dimer forms and reveal the intrinsic flexibility of the dimer. We also map the genetic mutations related to primary ciliary dyskinesia and asthenospermia on the head-neck complex. Moreover, we present the cryo-ET and sub-tomogram averaging map of mouse ependymal cilia and build the models for RS1-3, IDAs, and N-DRC. Contrary to the conserved RS structure, our cryo-ET map reveals the lack of IDA-b/c/e and the absence of Tektin filaments within the A-tubule of doublet microtubules in ependymal cilia compared with mammalian respiratory cilia and sperm flagella, further exemplifying the structural diversity of mammalian motile cilia. Our findings shed light on the stepwise mammalian RS assembly mechanism, the coordinated rigid and elastic RS-CP interaction modes beneficial for the regulation of asymmetric ciliary beating, and also facilitate understanding on the etiology of ciliary dyskinesia-related ciliopathies and on the ependymal cilia in the development of hydrocephalus.

Motile cilia or flagella are hair-like organelles functioning mainly as "paddles" to drive rapid cell movement or extracellular fluid flow. In vertebrates, motile cilia reside on the surface of epithelial cells in the ependyma, trachea, and fallopian tubes, whereas flagella are solely present in sperm. Dysfunctions in motile cilia and/or flagella lead to primary ciliary dyskinesia (PCD), which is characterized by chronic respiratory disease, organ misplacement, hydrocephalus, and infertility[1,2]. For instance, ependymal cilia, which line the surface of

[1]Key Laboratory of RNA Science and Engineering, Shanghai Institute of Biochemistry and Cell Biology, Center for Excellence in Molecular Cell Science, Chinese Academy of Sciences, University of Chinese Academy of Sciences, Shanghai 200031, China. [2]State Key Laboratory of Cell Biology, Shanghai Institute of Biochemistry and Cell Biology, Center for Excellence in Molecular Cell Science, Chinese Academy of Sciences, University of Chinese Academy of Sciences, Shanghai, China. [3]Ministry of Education-Shanghai Key Laboratory of Children's Environmental Health, Institute of Early Life Health, Xinhua Hospital, Shanghai Jiao Tong University School of Medicine, Shanghai 200092, China. [4]Shanghai Nanoport, Thermofisher Scientific, Shanghai, China. [5]National Facility for Protein Science in Shanghai, Shanghai Advanced Research Institute, Chinese Academy of Sciences, Shanghai 201210, China. [6]Key Laboratory of Systems Health Science of Zhejiang Province, School of Life Science, Hangzhou Institute for Advanced Study, University of Chinese Academy of Sciences, Hangzhou, China. [7]These authors contributed equally: Xueming Meng, Cong Xu, Jiawei Li, Benhua Qiu, Jiajun Luo. ✉e-mail: xlzhu@sibcb.ac.cn; yanx@shsmu.edu.cn; cong@sibcb.ac.cn

brain ventricle walls, create a unidirectional cerebrospinal fluid (CSF) flow through their coordinated back-and-forth beating. As CSF is rich in neuropeptides, its orderly flow is critical for nourishing the central nervous system and maintaining proper body axis. Moreover, it has been reported that improper function of ependymal cilia can lead to the obstruction of CSF flow, resulting in hydrocephalus in rodents, canines and zebrafish[3–5]. Recently, dysfunctions in ependymal motile cilia have been shown to lead to phenotypes resembling idiopathic scoliosis in zebrafish, suggesting a critical role of cilia-driven CSF flow in spine development[6,7].

The axoneme of motile cilia and flagella is generally composed of nine peripheral doublet microtubules (DMTs) surrounding a central pair (CP) of MTs (the "9 + 2" axoneme). Along the DMTs, seven inner dynein arms (IDAs), four outer dynein arms (ODAs), and three radial spokes (RSs) are arranged in 96-nm repeats to generate ciliary movement together with the CP[8,9]. The RS is a T-shaped protein complex with an orthogonal head pointing toward the CP and a stalk anchored on each A-tubule of the DMTs[8,10]. It acts as the mechanochemical transducer between the CP and axonemal dynein arms to regulate flagellar/ciliary motility[11–14]. The RS is a macromolecular machinery consisting of more than twenty subunits. The flagella of *Chlamydomonas*, a widely used protist model organism, contain two full-size RSs (RS1 and RS2) and a truncated RS3 with only the stalk in each 96-nm repeat unit of the axoneme. In contrast, motile cilia and flagella of most other protists, such as *Tetrahymena thermophile*, and metazoa possess triplet RSs (RS1 to RS3)[8,10]. Genetic mutations in the RS head-neck components also cause PCD[15–20].

During the past years, using cryo-electron tomography (cryo-ET) and sub-tomogram averaging, significant progress has been made in understanding the structures of RS in various organisms, including protists such as *Chlamydomonas*[8,9,13], *Tetrahymena*[8,21], *Choanoflagellate*[22], and *Trypanosoma*[23], as well as metazoa such as human respiratory cilia[18] and sea urchin[10,24], mouse[25,26], or human[25] sperm flagella. These efforts have led to a better understanding of morphologies, compositions, and variations of the RSs. We have previously determined the high-resolution cryo-EM structure of the mammalian RS head core complex[27]. Meanwhile, high-resolution cryo-EM structures of *Chlamydomonas* RS head and head-neck complexes have been reported[28,29]. These studies have revealed marked differences in morphology and composition between metazoan RSs and *Chlamydomonas* ones[8,10,22,27–35]. Interestingly, even within mammals, ciliary and flagellar RSs differ in morphology and composition[25–27].

In this work, we reconstitute a mouse RS head-neck complex and determine its near atomic resolution cryo-EM structures in both monomer and dimer forms. Our study reveals that the intrinsic flexibility of the RS head-neck dimer, combined with the proposed RS-CP contact modes, contribute to the elegant regulation mechanism of ciliary motility. We also provide potential etiology of RS head-neck gene mutations linked to PCD and asthenospermia[15,17,36–40]. Furthermore, we present the in situ axonemal structure of mouse ependymal cilia, including multi-scale models for RS1, RS2, RS3, IDAs, and N-DRC based on our cryo-EM and cryo-ET structures and AlphaFold2 predicted models. Altogether, our results shed light on the mammalian RS assembly mechanism, the mode of mammalian RS-CP interactions, and the structure-function relationships of ciliary and flagellar axonemes.

## Results

### Reconstitution of a stable mouse heteroheptameric RS head-neck complex

In a previous study, we successfully reconstituted the mouse RS head core complex, which consists of Rsph1-Rsph4a-Rsph9-Rsph3b subunits and exhibits a two-fold symmetric brake pad-shaped structure[27]. On this basis, we aimed to assemble and gain insights into a more complete mammalian RS head-neck complex. To determine whether the RS head and neck subunits can form a stable complex, we co-expressed Flag-tagged Rsph16 and *Strep*-Tag II-tagged Rsph1, together with Rsph3b, −4a, −9, −10b, −23, and −2 in HEK293F cells, and purified the complex by a two-step affinity purification strategy. SDS-PAGE followed by Coomassie brilliant blue staining indicated that a complex composed of Rsph1-Rsph4a-Rsph9-Rsph3b-Rsph23-Rsph2-Rsph16 was readily detected (Fig. 1a). The complex was further confirmed by glycerol-gradient ultracentrifugation and mass spectrometry analysis (Supplementary Fig. 1a, b).

### The RS head-neck complex exists as both monomer and dimer

Our cryo-EM analysis on the reconstituted mouse RS head-neck complex revealed the presence of both monomer and dimer forms (Fig. 1b and Supplementary Fig. 1c). We then determined the cryo-EM maps of the RS head-neck monomer and dimer at the consensus resolution of 3.28 Å and 3.57 Å, respectively (Fig. 1c, f, and Supplementary Figs. 2 and 3a–c). Furthermore, we conduced focused refinement on individual portions of the two maps, with the core region of the monomer reached 3.14 Å resolution, and the central bridge as well as the head regions of the dimer reached 3.36/3.46 Å resolution, respectively (Supplementary Figs. 2 and 3a, b, and Supplementary Table 1). Still, the relatively flexible neck region was resolved at lower resolution (discussed later). In this dataset, we also detected a less populated RS head core structure (Supplementary Fig. 2) that resembles the one obtained in our previous study[27] (Supplementary Fig. 3d). We then built atomic models for the RS head-neck monomer and dimer (Fig. 1d, g) based on the mouse RS head core structure[27] and AlphaFold2[41,42] predicted models for the neck subunits mostly based on the *Chlamydomonas* RS head-neck structures[28,29], with constraints of the current maps. The model fits in the corresponding map very well (Fig. 1c, f and Supplementary Movie 1) and displays high-resolution structural features (Fig. 1e, h).

In the monomer structure, the head portion consists of a compact body and two extending Rsph1 arms, made up of two copies of Rsph1/Rsph4a/Rsph9 (Fig. 1c, d), with the composition and conformation resembling that of the previous mouse RS head core structure (Supplementary Fig. 3f–g)[27]. Rsph3b is attached to one side of the head and is intimately associated and stabilized by the neck subunits Rsph2 and Rsph23, as well as the head subunits Rsph4a and Rsph4a' (Fig. 1d). This leads to a fluent connection between the head and the neck, which is in line with that observed in *Chlamydomonas*[28,29]. This RS head-neck associates laterally with one side of an arch bridge composed of two Rsph16 subunits to form the RS head-neck monomer (Fig. 1c, d). In contrast, the RS head-neck dimer consists of two copies of the RS head-neck connected by the arch bridge of Rsph16-Rsph16' (Fig. 1f, g).

We subsequently conducted cross-linking and mass spectrometry (XL-MS) analysis on the RS head-neck complex by using the amine-to-amine crosslinker BS3 (bis (sulfosuccinimidyl) suberate), which has a spacer-arm-length of 11.4 Å[43,44]. The result overall supports the subunit spatial arrangement determined by the cryo-EM study (Fig. 1i and Supplementary Table 3). Notably, although Rsph10b appeared to be moderately abundant in the affinity-purified reconstituted RS head-neck complex (Supplementary Fig. 1b), it was only weakly detected by XL-MS (Fig. 1i) and completely absent in the RS1/RS2 head-neck maps (Fig. 1c, f) after subjecting the sample to an additional round of glycerol gradient centrifugation. Such results suggest that Rsph10b and some of the head-neck subunits may form a complex distinct from the head-neck complex of RS1/RS2.

### Subunit interaction network of the RS head-neck complex

The current RS head-neck structures disclosed additional interactions in the head-neck interface and within the neck (Fig. 2a and Supplementary Table 4), along with that within the head[27]. The Rsph4a N-terminal Dpy-30 motif extends to attach underneath the head subunit Rsph9 through rich H-bonds/salt-bridges and also clip the neck subunit Rsph3b (Fig. 2a and Supplementary Fig. 4a). Indeed, the

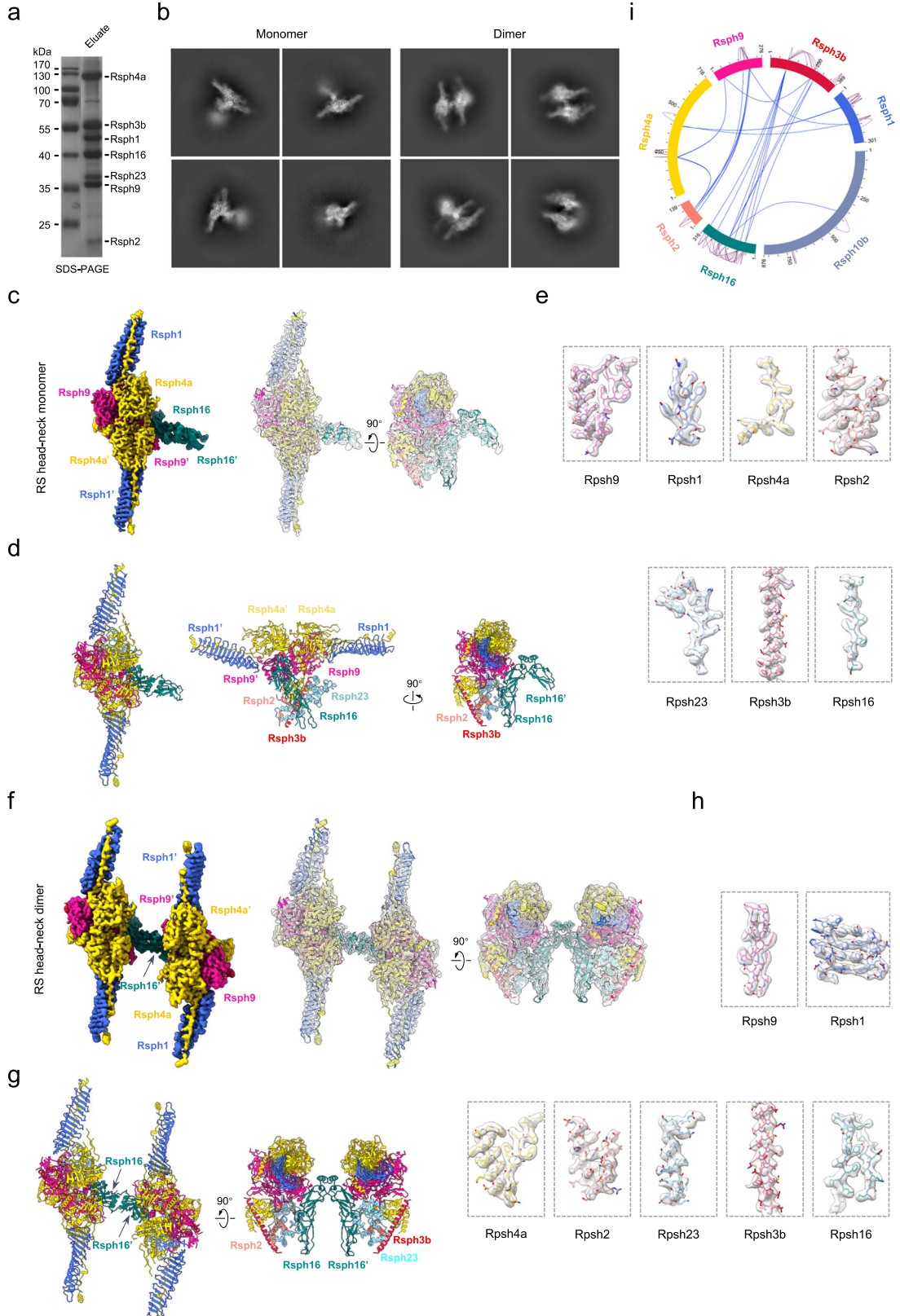

**Fig. 1 | Cryo-EM structures of reconstituted mouse RS head-neck complex.**
**a** Coomassie-stained PAGE of the reconstituted RS head-neck complex. This was repeated three times independently with similar results. Source data are provided as a Source Data file. **b** Representative reference-free 2D class averages of the RS head-neck complex, showing the existence of monomer and dimer. **c** Cryo-EM map of the RS head-neck monomer and model-map fitting, with subunits shown in distinct colors. This color schema was followed throughout. **d** Atomic model for the RS head-neck monomer. **e** Representative high-resolution structural features for

each of the individual subunits of the monomer. **f** Cryo-EM map of the RS head-neck dimer and the model-map fitting. **g** Atomic model for the RS head-neck dimer. **h** Representative high-resolution structural features for each of the individual subunits of the dimer. **i** XL-MS analysis of the RS head-neck complex. Identified cross-linked inter-subunit contacts are shown as blue lines, and intra-subunit contacts as purple lines. We used best E-value (1.00E−02) and spec count of at least 2 as the threshold to remove extra XL-MS data with lower confidence.

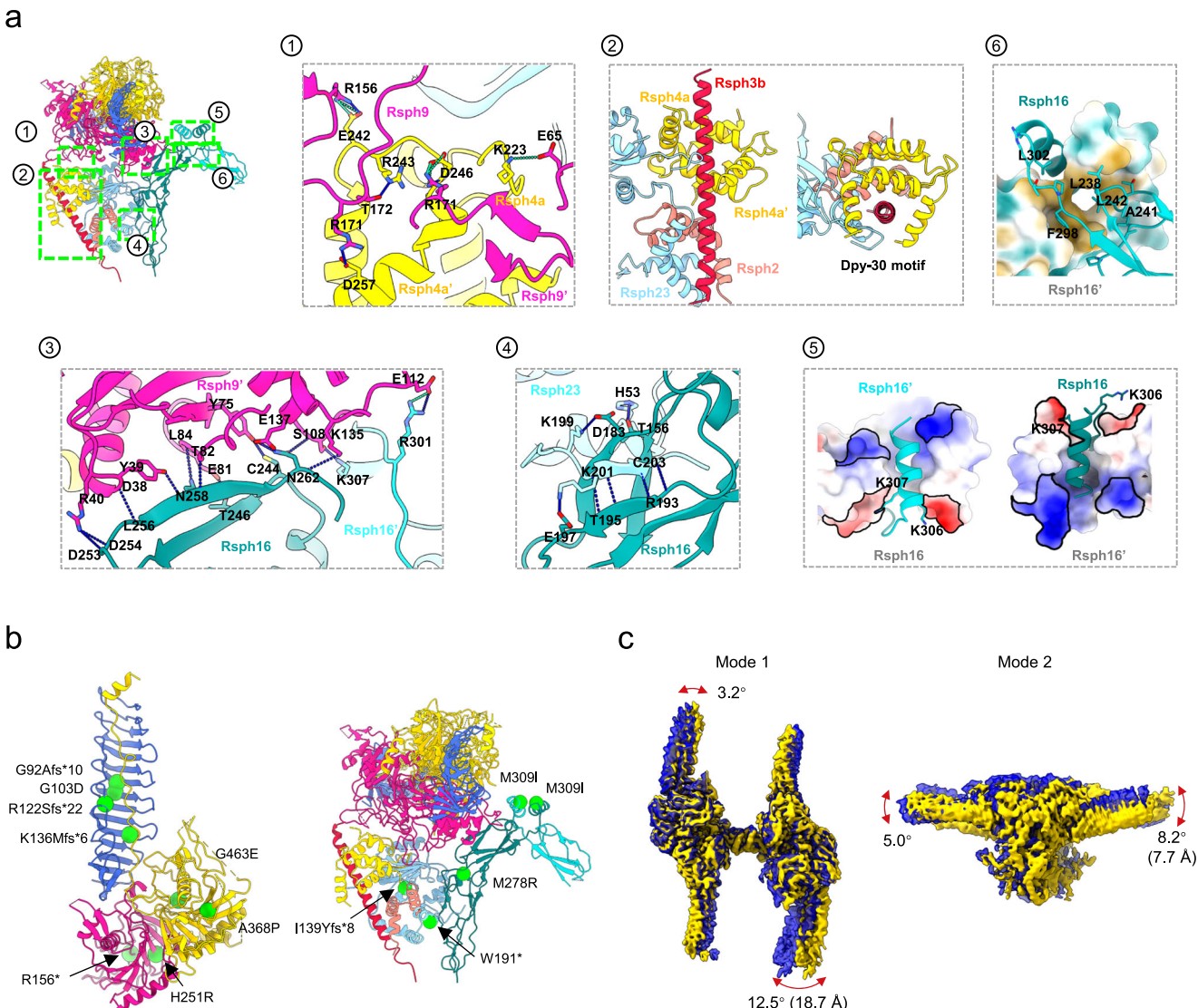

**Fig. 2 | Subunit interaction networks and mapping of the PCD/asthenospermia-related mutations on the RS head-neck complex, as well as the intrinsic flexibility of the head-neck dimer. a 1-** The interaction networks of Rsph9-Rsph4a; **2-** between Rsph3b and the Dpy-30 motifs of Rsph4a/Rsph4a'/Rsph2/Rsph23; **3-** Rsph9-Rsph16; **4-** Rsph23-Rsph16. **5-** The electrostatic interaction between the C-terminal a-helix of Rsph16 and Rsph16'; **6-** and the hydrophobic interaction between the Rsph16 and Rsph16'. H-bonds are drawn in blue dash line and salt bridges in green dash line. **b** Known PCD/asthenospermia disease-causing mutations mapped onto the model of the RS head-neck complex. Each green ball represents a mutation associated with PCD/asthenospermia disease. **c** Two representative modes of motion of the RS head-neck dimer. The angular range and direction as well as the extreme distance of the motion are displayed on the overlaid two extreme maps in the motion (in blue and yellow).

Dpy-30 motifs from not only the head subunits Rsph4a and Rsph4a' but also the neck subunits Rsph2 and Rsph23 wrap around the long α-helix of Rsph3b (Fig. 2a), so that Rsph3b intimately engages the head and the neck subunits. The RS head-neck is as well connected with the arch bridge through an intense H-bond/salt-bridge interaction network between Rsph9 and Rsph16 (Fig. 2a) and H-bond interactions between Rsph23 and Rsph16 (Fig. 2a). In addition, in the arch bridge, the Rsph16-Rsph16' interaction appears to occur in two areas, involving electrostatic interactions between their C-terminal α-helix (Fig. 2a) and hydrophobic interactions between a part of the DNAJ-C domain (Fig. 2a and Supplementary Fig. 4a). Interestingly, Rsph16 is a homologue of HSP40, a subtype of DnaJ cochaperone[45]. As in many DnaJ family proteins, Rsph16 also uses the C-terminus region for dimerization[46].

Mammalian RS1/RS2 head appears to be much more reduced in composition and morphology than the *Chlamydomonas* RS head[28,29,34] (Supplementary Fig. 4b, c), despite the extensive homologies in structural motifs and sequence similarities among orthologs throughout evolution (Supplementary Fig. 4a). In *Chlamydomonas* RS head, RSP4/RSP6 cover the head surface and RSP1/RSP10 form the asymmetric two arms. In contrast, the mammalian RS head has two copies of Rsph4a on the head surface and two copies of Rsph1 forming the symmetric arms; whereas the mouse Rsph6a is specifically expressed in sperm for flagellar formation[17,47]; and our previous[27] and the current (Fig. 1c, f) data do not suggest Rsph10b as a component of mammalian RS1/RS2 head. Moreover, the *Chlamydomonas* RS head-neck has an additional subunit RSP5 and a much longer C-terminal portion of RSP2 that extends to the head surface[28,29], which are all absent in mammalian RS head-neck complex.

## Mapping of PCD/asthenospermia-related mutations on the RS head-neck complex

Since many pathogenic mutations of PCD or asthenospermia patients on the RS head-neck subunits have been documented (Supplementary Table 5)[15,17,36,38–40,48], our atomic model of the head-neck complex

enabled us to map these mutations on this structure (Fig. 2b). Frameshift mutations such as G92Afs*10, R122Sfs*22, and K136Mfs*6 in the MORN motif of Rsph1[17] truncate the protein and might also disrupt its interaction with Rsph4a, thus disturbing the arm formation of the RS head (Fig. 2b). Similarly, the W191* mutation of Rsph23[39] might disrupt the head-neck assembly by truncating the protein (Fig. 2b). The I139Yfs*8 mutation of Rsph23 truncate its C-terminus[40], which is responsible for its binding to Rsph16 to stabilize the head-neck complex, and may therefore induce instability of the complex.

Missense mutations in Rsph16, such as M278R and M309I, have been reported to be related to male fertility in asthenozoospermia[15,38]. The M278R mutation changes the hydrophobic Met into a hydrophilic Arg with a longer sidechain. This change may result in a potential clash and repulsion with the adjacent R225, which could be propagated to R189 located on the other side of R225 through H-bond networks (Supplementary Fig. 4d). R189 resides in the loop involved in the interaction of Rsph16 with the C-terminal tail of Rsph23, thus eventually disturbing the interaction of Rsph16 with Rsph23 and weakening the stability of the head-neck complex. The M309I mutation is located at the Rsph16 dimerization interface formed by the C-terminal α-helix (Fig. 2b). The mutation of Met into Ile may disturb the formation of the arch bridge and impair the stability or even formation of the head-neck dimer.

## Conformational flexibility of the RS head-neck dimer

Our local resolution analysis of the RS head-neck dimer revealed relatively lower local resolution at the very end of the Rsph1 arm and the peripheral portion of the neck (Supplementary Fig. 3b). These regions also appear "fuzzy" in the reference-free 2D class averages (Fig. 1b). These observations indicate that these less constraint regions exhibit plasticity. To further examine the intrinsic flexibility of the complex, we performed 3D flexible refinement (3DFlex)[49] on the head-neck dimer in cryoSPARC and found considerable relative movements between the two monomers (Fig. 2c). Specifically, Mode 1 obtained through this analysis mainly describes an open-and-close movement between the Rsph1 arms of the two monomers with an angular range of Rsph1-rotation up to 12.5° (Fig. 2c and Supplementary Movie 2). Mode 2 corresponds more to the seesaw-like alternating up-and-down motions between the two monomers, with an angular range of Rsph1 arm rotation up to 8.2° (Fig. 2c and Supplementary Movie 3). These structural variations imply flexibility that may allow RS to transduce forces between CP and DMTs as cilia in natural environments bend or twist, not limited to rhythmic beating with identical waveform and beat frequency. This flexibility may potentially facilitate the formation of asymmetric planar beat patterns of cilia.

## Cryo-ET maps of RS1 and RS2 in ependymal cilia exhibit good match with the RS head-neck dimer cryo-EM map

To clarify whether the RS head-neck dimer is a bona fide one in motile cilia, we determined the in situ axonemal structure of ependymal cilia in cultured mouse ependymal cells (mEPCs), using cryo-ET combined with sub-tomogram averaging, and achieved a consensus map at the resolution of 24.2 Å (Fig. 3a, Supplementary Fig. 5a, and Supplementary Table 2). We performed further focused refinement on the RS1/RS2/RS3 region to improve structural details (Supplementary Fig. 5b and Supplementary Movie 4). We then fitted our high-resolution cryo-EM map of the reconstituted RS head-neck dimer into the cryo-ET map of RS2 in the head-neck region, and found that they overall match each other very well especially in the head region, and the lower portion of the neck may potentially contain additional component (Fig. 3b). This is also the case for the in situ RS1 map in the head-neck region (Fig. 3c). Collectively, these data suggest that our recombinant RS head-neck complex represents the endogenous head-neck architecture of both RS1 and RS2, thus further confirming that the mammalian RS1/RS2

head-neck is simplified in composition compared to its counterpart in *Chlamydomonas*[8,27–29,32].

## Structural models of mammalian RS1/RS2/RS3

To achieve a complete model of the mammalian RS1 and RS2, we first fitted our RS head-neck dimer model into the corresponding region in our RS1/RS2 cryo-ET maps of ependymal cilia. We then fitted AlphaFold2 predicted models of several RS stalk/base components into their corresponding locations, based on features of the cryo-ET map and models, as well as previous reports on mammalian[50] and *Chlamydomonas* RSs[28,32]. For RS1, in addition to the head and neck region, we fitted the models of Cyb5d1, Rsph12, Rsph11, Rsph14, Rsph20, Rsph22, and IQUB in the stalk and base regions (Fig. 3d and Supplementary Table 6). For RS2, we also fitted the Cyb5d1, Rsph12, Rsph11, Rsph14, Rsph15, Rsph22, and MORN3 subunits in the stalk and base regions (Fig. 3e).

It has been reported that stalk and base regions of *Chlamydomonas* RS1 and RS2 respectively contain 8 and 12 copies of RSP22[28]. While we similarly incorporated 12 copies of Rsph22 into the mammalian RS2 (Fig. 3e), 8 copies of Rsph22 still failed to occupy the extra density of the RS1 (Fig. 3d). Based on the good fit between the model and the map as well as our MS analysis (Supplementary Table 7), we placed one copy of adenylate kinase 8 (AK8), a nucleoside monophosphate kinase associated with congenital hydrocephalus in knockout mice[51], in the stalk region of RS1 together with five copies of Rsph22 (Fig. 3d). To verify this assignment, we first examined the subcellular localization of GFP-AK8 in cultured mEPCs by super-resolution fluorescence microscopy and confirmed that GFP-AK8 co-localizes with Rsph4a, an RS marker, in the axonemal central lumen of ependymal cilia (Supplementary Fig. 7a). Secondly, as our model (Fig. 3d) implicated an interaction between Rsph3b and AK8, we coexpressed FLAG-Rsph3b and GFP-AK8 in HEK293T cells and readily detected an association between the two proteins through co-immunoprecipitations (CoIP) (Supplementary Fig. 7b).

Noteworthily, the composition and the high-resolution structure of RS3 remain largely unknown. To provide structural insights into this unique RS, we managed to fit several potential components into our RS3 cryo-ET map based on its structural features and recent reports on the potential composition of mammalian RS3[20,52,53]. For instance, LRRC23 is a potential RS3 component because its absence abolished the RS3 head in mice and humans[20]. Indeed, the AlphaFold2 predicted model of LRRC23, which displays a characteristic curved feature, matches a bent density underneath the RS3 head very well (Fig. 3f). In line with recent reports suggesting that LRRC23 contacts with Rsph9[20,53], we observed a head region right above LRRC23 displaying an obvious double hump feature, which matches the core of the RS head (Rsph4a/Rsph9) reasonably well (Fig. 3f). In addition, a previous cryo-ET study on trachea cilia of *Rsph4a*-deficient mice indicated the loss of all triplet RS heads[54], and another study on respiratory cilia from *Rsph1*- or *Rsph4a*-deficient PCD patients also suggested their involvement in RS3[55]. Collectively, these results indicated that the head subunits Rsph4a/Rsph9/Rsph1 could also exist in RS3. We then fitted a RS head core complex, with one Rsph1 arm substituted by a Rsph10b, into the lower half of the RS3 head density (Fig. 3f). In this proposed fitting, the C-terminal globular domain of Rsph10b fits well in a hook-like density, and the N-terminal Dpy-30 motif of two Rsph4a subunits held in the wing-like density reasonably well (Fig. 3f). Moreover, in the other half of the RS3 head, a head core containing two Rsph9 and one Rsph4a could be fitted into a hump-like region (Fig. 3f), and the long loop of Rsph4a could snugly fit into the MORN motif of another Rsph10b, whose C-terminal globular domain matches well with another hook-like density extending towards the adjacent RS2 (Fig. 3f).

Moreover, previous cryo-ET study has demonstrated that *Tetrahymena* Cfap61 and Cfap251 locate in the central stem region and the base of RS3, respectively[56]. We then fitted mouse Cfap61 and Cfap251,

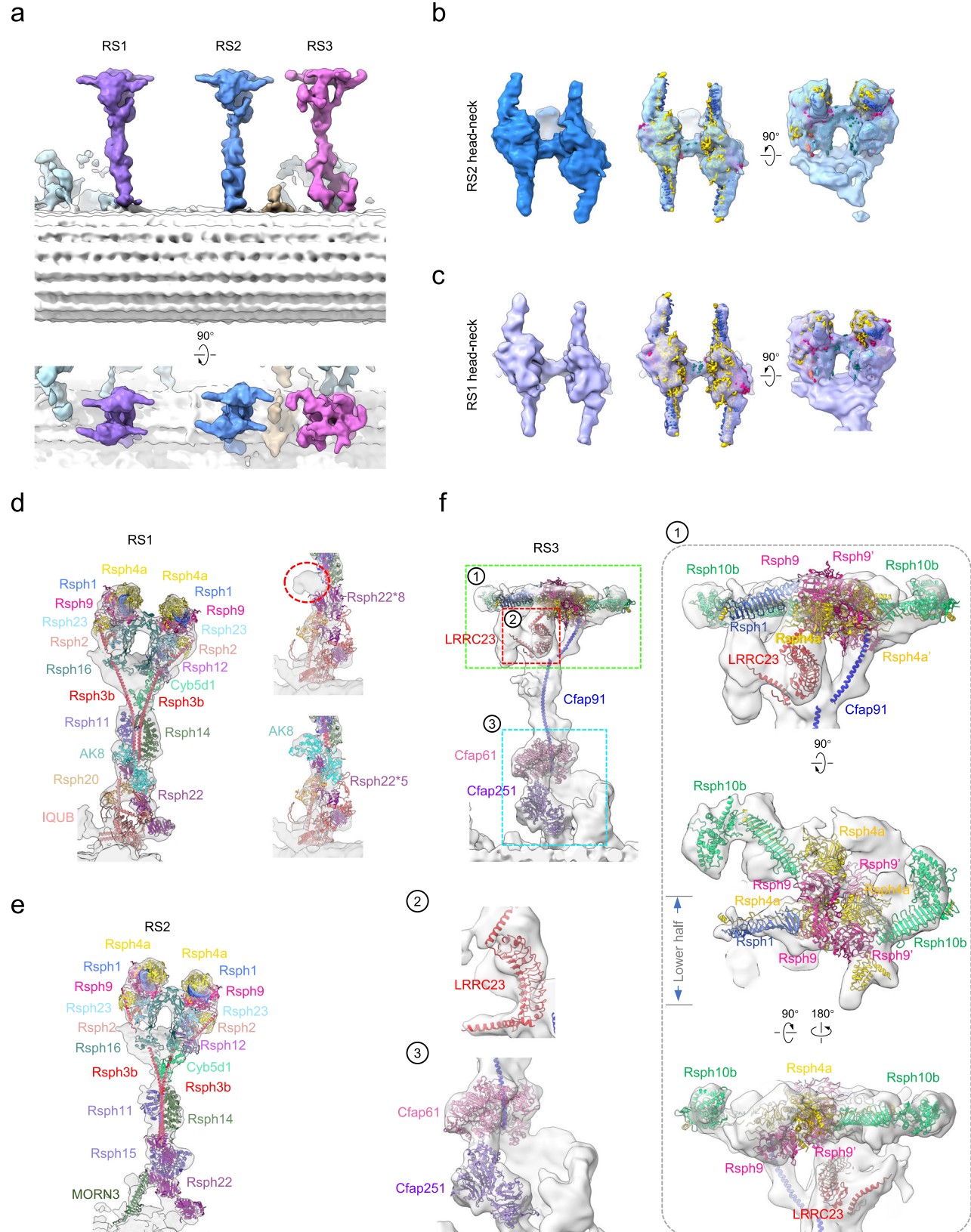

**Fig. 3 | The in situ cryo-ET structure of the mouse ependymal cilia axoneme and the proposed multi-scale models for RS1/RS2/RS3. a** Different views of the cryo-ET map of the 96-nm axonemal repeat of mouse ependymal cilia, with RS1/RS2/RS3 shown in distinct color. Fitting of our cryo-EM map of the RS head-neck dimer (in color) into the corresponding head-neck region of the in situ cryo-ET map of RS2 (**b**, blue) or RS1 (**c**, purple), indicating a good matching between them. **d** A proposed model of mammalian RS1, with our RS head-neck dimer model and AlphaFold2 predict models for the stalk/base fit into the cryo-ET map of RS1. While the stalk and base region of *Chlamydomonas* RS1 is reported to contain 8 copies of RSP22, we propose that 1 copy of AK8 and 5 copies of Rsph22 fit better than 8 copies of Rsph22 in the stalk and base region of mammalian RS1 (right panels). **e** A proposed model of mammalian RS2. **f** A proposed model for the mammalian RS3. Representative RS3 head, neck, and base regions are displayed with magnified views (**1**- head, **2**- neck, **3**- base regions) to show details.

respectively, into similar locations in the stem and base of RS3, which matches the structural features well (Fig. 3f). Furthermore, since *Tetrahymena* Cfap91 is reported to be an essential component of RS3 critical for the localizations of Cfap61 and Cfap251[52], we placed mouse Cfap91 in the corresponding root region of RS3, with its long α-helix extending dramatically upwards to the RS3 head (Fig. 3f)[20,53,56,57].

Furthermore, our focused 3D classification on the RS1/RS2/RS3 region, respectively, revealed two RS1 conformations relative to the DMT (Fig. 4d), while RS2/RS3 remained relatively stable. Specifically, in one state, RS1 stands more perpendicular to the underneath DMT, while in the other state, RS1 appears more tilted (about 7.6°) towards the adjacent RS3 from the next 96-nm repeat. Distribution analysis showed that the population of the tilted RS1s is approximately 1.6 times of the perpendicular ones, and it appears that the perpendicular RS1s are scattered as small clusters among the tilted RS1s (Fig. 4d). We also observed the occasional downward tilt of a complete RS1 (Supplementary Fig. 5c). Taken together, RS1 might be more dynamic, exhibiting larger conformational space with RS1 in perpendicular, tilted, and occasionally downward tilted orientations, likely due to the lack of constraint from its neighboring RSs. In contrast, RS2 and RS3 are mostly perpendicular and relatively stable, which could be attributed to the mutual contact in their head region in mammalian cilia[16,18]. We then postulate that RS1 may be more dynamic in regulating ependymal cilia motion.

### Reduced IDAs and MIPs in DMT of ependymal axoneme

Although both ependymal cilia and tracheal cilia exhibit a coordinated metachronal waveform, they differ in their beat frequencies. Ependymal cilia beat at a frequency of around 34 Hz at 37 °C, whereas tracheal cilia at approximately 19 Hz[58]. Sperm motility arises from the beating of a single flagellum capable of a variety of waveforms, depending on species, activation state, and environments, at a frequency of up to 20 Hz[59]. Even though cryo-ET maps of mammalian respiratory cilia and sperm flagella are available[18,25,26,54], to the best of our knowledge, no cryo-ET or cryo-EM structure has been reported for ependymal ciliary axonemes to date. To facilitate a direct comparison among axonemes of different tissues for potential tissue-specific structural variations, we performed further focused refinement on the corresponding regions of the consensus map and acquired in situ maps of DMT, IDAs, and the nexin-dynein regulatory complex (N-DRC) in ependymal cilia at the resolution of 19.6 Å for DMT and ~24.0 Å for the other two (Fig. 4a–c, Supplementary Figs. 5b and 8d, and Supplementary Table 2).

Noteworthy, compared with the cryo-ET maps of human respiratory cilia[18] (Fig. 4a) and mouse sperm flagella[25] (Supplementary Fig. 8a), IDA-b/c/e were all missing in ependymal cilia. In addition, IDA-a density is relatively weak (Fig. 4a), and through further 3D classification we found a class with nearly no IDA-a density (Fig. 4e), indicating a variation of IDA-a in different structural repeats. We also managed to fit in the density of different IDAs with AlphaFold2[41,42] predicted models of Dnah1, Dnah2, Dnah6, Dnah10, Dnah12, Mbl1, and Mbl2 in the density of IDAs (Fig. 4b) by referring to the homologs in *T. thermophila*[52], *Chlamydomonas*[60], and evolutionary analysis[61]. Accordingly, we confirmed that Dnah12 is located in IDA-a, Dnah1 in IDA-d, Dnah6 in IDA-g, and Dnah10/2 in IDA-I1/fα/fβ (Fig. 4b). Besides, Mbl1 and Mbl2 are located in the stalk of IDA-d as a hetero-dimer (Fig. 4b).

Axonemal microtubule inner proteins (MIPs) associate with the inner surface of the microtubule wall, which are important for microtubule stability and structure. Interestingly, we found that MIPs are also dramatically reduced in mouse ependymal cilia compared with bovine/human respiratory cilia and mouse sperm flagella (Fig. 4c)[18,25,62]. Specifically, Tektin1-4 and Tektin bundle interacting protein 1 (TEK-TIP1) within the A-tubule of DMT[25] are all missing in mouse ependymal cilia (Fig. 4c). Similar to ependymal cilia, Tektins are also absent in the A-tubule of *Chlamydomonas* DMTs (Supplementary Fig. 8b)[63,64]. In contrast to the mouse ependymal cilia, mouse sperm flagella exhibit

many more filamentous MIPs, such as the Tektin bundle and other sperm-specific filaments inside the A-tubule and striated densities attached to the inner wall of the B-tubule[25] (Fig. 4c).

Collectively, these results demonstrate significant tissue specificity for both IDAs on and MIPs within DMT, which may be related to distinct external environments these cilia are facing and also to their distinct pattern of motion.

### RS-CP contact patterns

To gain further insights into the RS-CP contact patterns, we incorporated our mouse RS head-neck fitted DMT of mouse ependymal cilia and the cryo-ET map of mouse sperm CP (EMD-27445 https://www.ebi.ac.uk/emdb/EMD-27445)[25] into a "9 + 2" axoneme by matching remaining RS densities of the CP (Fig. 5a). This fitted multi-scale axoneme structure reveals apparent complementary geometries between the CP projections and the RS heads: (1) CP projection 1d appears to hold the Rsph4a teeth in both RS heads of the DMT8 head-neck dimer, exhibiting a rigid contact mode when they contact during ciliary beat (Fig. 5b, c); (2) CP projections 2b/2d contact the Rsph1 arms from the DMT4 head-neck dimer (Fig. 5b, c), which could lead to elastic RS-CP contact given the intrinsic open-and-close and seesaw-like alternating up-and-down motions happened mainly in the Rsph1 arms of the head-neck dimer (Fig. 2c); (3) CP projection 2d contacts both Rsph1 arms of one RS head and the Rsph4a teeth of the other head of DMT3 (Fig. 5b), exhibiting a combined elastic-and-rigid interaction mode of RS-CP.

Moreover, our analysis of the surface properties of the RS head-neck suggested an overall negative electrostatic potential on the CP-facing RS head region, especially for Rsph4a, and a generally hydrophilic property in this region (Fig. 5d). The complicated geometries and surface properties of projections at and around each presumed contact site might accordingly facilitate the formation of asymmetric planar beat patterns of cilia[12].

## Discussion

In the present study, we determined the near atomic resolution cryo-EM structures of the mammalian RS head-neck complex (Fig. 1). Our structural analysis revealed a complex subunit interaction network and intrinsic modes of motion within this regulatory machinery of ciliary motility (Fig. 2a, c). We also identified the potential etiology of RS head-neck gene mutations that have been linked to PCD and asthenospermia (Fig. 2b)[15,17,36,38–40,48]. Moreover, we presented the cryo-ET axoneme structure of ependymal cilia (Fig. 3), revealing its distinct morphology, especially reduced IDAs on and MIPs within DMT, compared with that of mammalian respiratory cilia and sperm flagella (Fig. 4a, c). In mouse ependymal and mammalian respiratory cilia, multiple sperm-specific component densities connecting RS1/RS2/RS3 were absent (Supplementary Fig. 8c)[18,25]. These components are likely trimmed during evolution for their divergent environments and modes of motion. Moreover, based on our cryo-EM and cryo-ET structures, we proposed a structural model for mammalian RS3, especially for its head-neck, and built more complete multi-scale models for the RS1/RS2, IDAs, and N-DRC of ependymal cilia (Figs. 3d–f and 4b, and Supplementary Fig. 8d). We also identified an extra subunit, AK8, located in the stalk of RS1, and proposed a stepwise assembly mechanism of the mammalian RS1/RS2 head-neck complex. Collectively, our study sheds lights on the tissue specificity and evolution of mammalian cilia among ependymal and respiratory cilia and flagella, the complex mode of mammalian RS-CP interaction (Fig. 5a–d), and how these factors collectively regulate ependymal cilia motility.

Based on our reconstitution and cryo-EM study on mammalian RS head-neck complex and the multiple intermediate states obtained from one dataset, we propose a stepwise assembly mechanism of the RS head-neck complex for RS1/RS2 (Fig. 5e): (1) the RS head, consisting of two Rsph4a-Rsph9-Rsph1 protomers, can be assembled

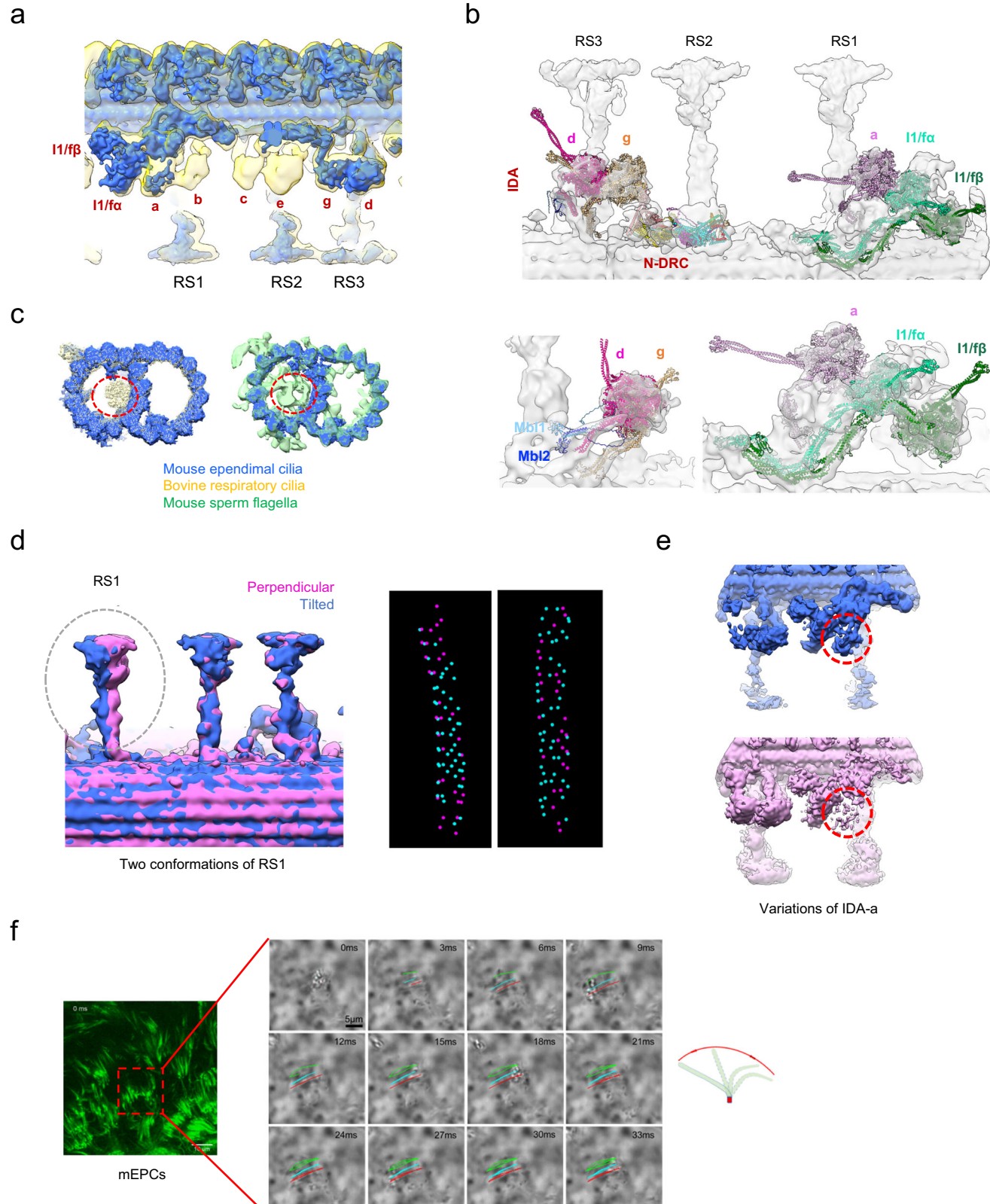

**Fig. 4 | Reduced IDAs and MIPs in DMT of mouse ependymal axoneme revealed by cryo-ET. a** Overlaid cryo-ET maps of mouse ependymal cilia axoneme (royal blue) with human respiratory ciliary axoneme (transparent yellow, EMD-5950) revealed that IDA-b/c/e are absent in mouse ependymal cilia. **b** Proposed models for the IDAs of mouse ependymal cilia. **c** Cross-section view of the overlaid cryo-ET maps of the mouse ependymal cilia DMT 48-nm repeat (royal blue) vs. the bovine respiratory DMT (transparent khaki, EMD-24664) or mouse sperm flagella (transparent green, EMD-27455), revealed greatly reduced MIPs (e.g., missing Tektins) within the A-tubule of ependymal cilia DMT (indicated by dotted red circles). **d** 3D classification revealed two in situ conformations of RS1 (left) and the in situ distribution of the two conformations in two representative tomograms (right). The perpendicular RS1 conformation is shown in magenta, and the tilted one in blue. **e** Focused 3D classification revealed two in situ IDA-a states (indicated by dotted red circle), in one of which the IDA-a appears nearly missing (lower panel). **f** The beating pattern of mouse ependymal cilia (repeated three times independently with similar results).

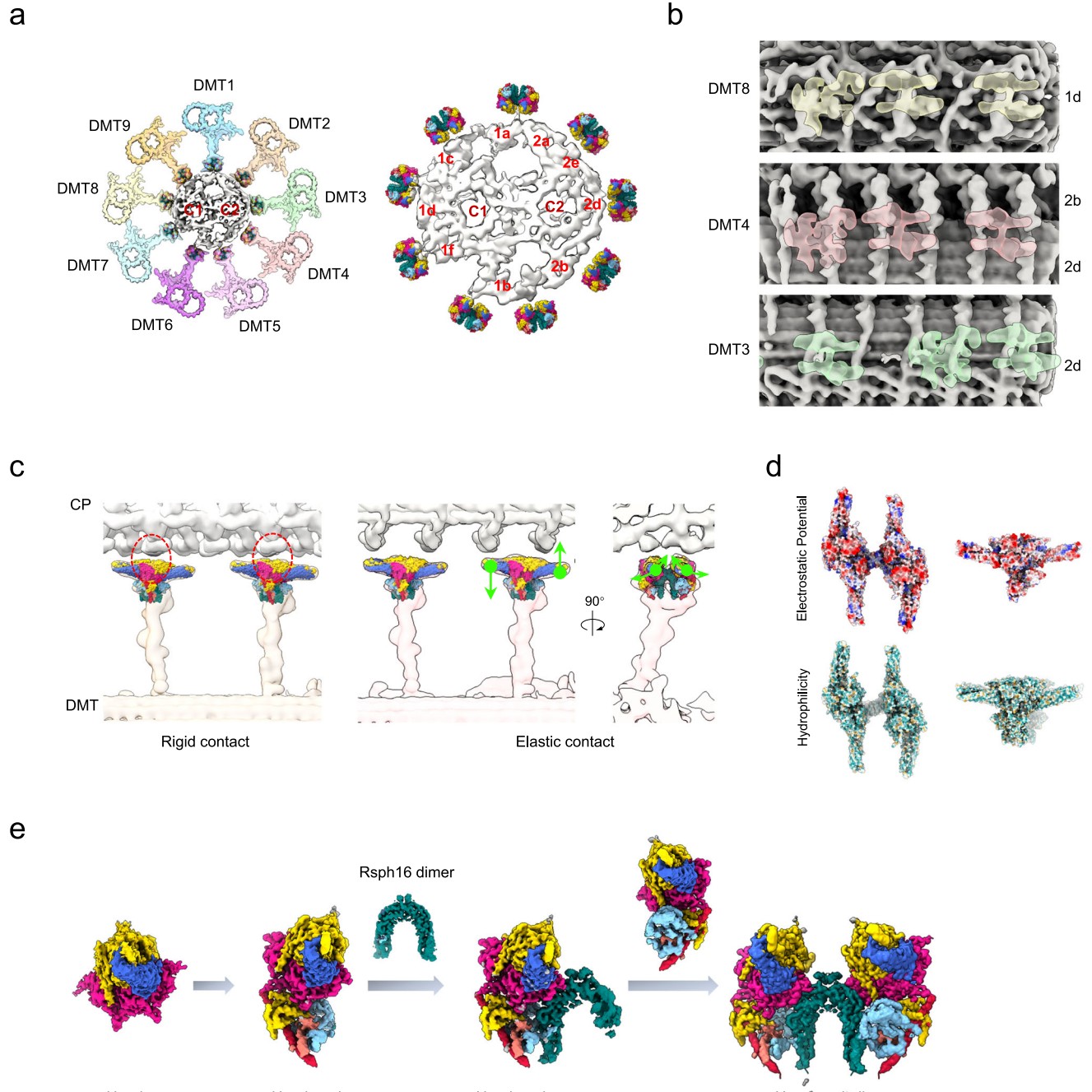

**Fig. 5 | Proposed RS-CP interaction patterns and the assembly mechanism for mammalian RS head-neck complex. a** Coordination of the RS head-neck complex-fitted DMT of our mouse ependymal cilia and the recent cryo-ET map of mouse sperm CP (EMD-27445) into the framework of a 9 + 2 axoneme by matching remaining RS densities of the CP. **b** Longitudinal views of locations of the RSs (transparent density) relative to the indicated periodic CP projections (in grey). The representative RSs from DMT3/4/8, better matched with the remaining RS densities of the CP, are shown here. **c** Proposed RS-CP interaction patterns, in coordination with the intrinsic flexibility of RS head-neck complex. Left: rigid contact. Right: elastic contact, incorporation of the seesaw-like alternating up-and-down motions (left) and open-and-close motion (right). **d** The surface properties of the RS head-neck dimer. Coulombic electrostatic potential, coloring ranging from red (negative) to blue (positive). Hydrophilicity, coloring ranging from dark cyan (most hydrophilic) to dark goldenrod (most lipophilic). **e** The proposed assembly mechanism for the mammalian RS head-neck complex of RS1/RS2.

independent of other subunits; (2) the neck subunit Rsph3b appends to one side of the head and then be wrapped around by the Dpy-30 motif of the head subunits Rsph4a/Rsph4a' and the neck subunits Rsph2/Rsph23, connecting the head and neck together; (3) Rsph16 can form a homodimer independent of other subunits, since the Rsph16 homodimer appears already present in our head-neck monomer structure and free Rsph16 homodimer has also been detected in previous biochemical study[45]; (4) one RS head-neck attaches on one side

of the arch bridge formed by the Rsph16 dimer, forming the head-neck monomer; (5) then another head-neck engages the other side of the arch bridge, forming the RS head-neck dimer. Rsph16, the homologue of HSP40 and also comprising a C-terminal dimerization domain[46], appears to play an essential role in the dimerization of the RS head-neck complex (Fig. 1). Since the reconstituted RS head-neck complex can dimerize (Fig. 1), the RS head-neck dimer might be similarly pre-assembled in mammalian cell bodies. The dimer might then be

transported into cilia for incorporation into the stalk/base portion, which can be assembled independently of the dimer on DMTs[18], through the long α-helix of Rsph3b (Fig. 3d, e). Despite these, detailed assembly mechanisms of the complete RS1/RS2 require future investigations.

We also proposed a model for mammalian RS3 based on the structural features of the map and model, as well as available biochemical and genetic results (Fig. 3f)[20,52–56]. The model provides a reliable fit for Cfap251/Cfap61/Cfap91 in the stalk and base region, LRRC23 in the neck, and Rsph9/Rsph4a/Rsph1 in the head of RS3. The fitting of Rsph10b is also reasonable, as its characteristic hook-like feature of the C-terminal globular domain matches the corresponding densities well. In this arrangement, the RS3 head can hold about two RS head complexes, with some extra densities remaining to be filled in (Fig. 3f). Its slightly larger size than that of RS1/RS2 also suggests a more complex composition and a different role in the RS-CP interaction and cilia motion regulation. The proposed model for RS3, however, still requires future validation, for instance, through a high-resolution structure of mammalian RS3.

Furthermore, our multi-scale analysis of the RS-CP interaction revealed the rigid contact mode between RS and CP, in addition to the elastic and rigid-elastic RS-CP contact modes (Fig. 5b, c). These rigid RS-CP contacts could constrain the relative movement between CP and RS, which combined with the elastic and rigid-elastic RS-CP contacts could transfer the mechanical force generated by dynein-induced DMT sliding motion[24] to CP and form constraints between DMT and CP. These factors could potentially mediate the cilia mode of motion, collectively generating an asymmetric planar beat pattern of cilia.

Although flagella and many motile cilia share a common feature of the "9 + 2" MT arrangement, they exhibit different beating patterns and frequencies[65]. For instance, sperm flagella generally beat in a whip-like motion along a helical path, whereas motile cilia, including tracheal and ependymal cilia, beat in a back-and-forth manner (Fig. 4f). Ependymal cilia are generally longer than tracheal cilia and beat at a faster rate[58]. Related to this, our cryo-ET axoneme structure of mouse ependymal cilia revealed simplified MIPs of DMT without the bundle of Tektins in the A-tubule, which differs from that of mammalian respiratory cilia and sperm flagella (Fig. 4c)[25,62]. Since Tektins form hyperstable polymers[66], the absence of Tektin bundles within the A-tubule of DMT in ependymal cilia may make them structurally less stiff and stable. Respiratory cilia, ependymal cilia, and sperm flagella function in distinct physiological and environmental contexts. While respiratory cilia and sperm flagella must move the sticky mucus or swim up through the female reproductive tract to reach the egg, ependymal cilia facilitate the circulation of water-like CSF. Collectively, we propose that, to handle the less viscous environment of CSF, ependymal cilia may not require Tektins within the A-tubule of DMT to provide extra stabilization/stiffness of DMT. Substantiating our notion, *Chlamydomonas* flagella, which mostly swim in the water, also lacks the Tektin bundle within the A-tubule of its DMT (Supplementary Fig. 8b)[64].

The axonemal dynein on DMTs, especially the IDAs which determine the waveform of ciliary/flagellar motion, produces force that is spatiotemporally regulated[67]. Interestingly, our cryo-ET structure revealed a significant reduction in IDAs in ependymal cilia. Specifically, the consecutive IDA-b/c/e in all DMTs are absent, and part of the IDA-a is also missing (Fig. 4a, e). Although it has been reported that IDA-b/e are missing from at least two DMTs in mouse respiratory cilia[68], and one of the DMTs lacks IDA-b in *Chlamydomonas*[69,70], the complete loss of IDA-b/c/e in all DMTs has not been observed in any other tissues. This indicates that this is a unique ependymal tissue-specific feature. Moreover, it has been reported that a *Chlamydomonas* mutant lacking IDA-c shows only a slight swimming defect in normal culture medium but a greatly reduced swimming ability in viscous media[71]. Therefore, we postulate that IDA-b/c/e may mainly contribute to the force generation in viscous environments. Accordingly, their natural loss in

mammalian ependymal cilia is attributed to an adaption to the watery low viscosity CSF during evolution. Alternatively, the absence of certain key structures might result from cell separation or cryovitrification. It is currently difficult to definitively exclude this possibility.

It is also interesting to note that, from mammalian sperm flagella to respiratory cilia then to ependymal cilia, ciliary composition and ultrastructure become increasingly simplified (Fig. 4a–c and Supplementary Fig. 8a–c)[18,25]. These simplifications might be related to the differences in their working environments and functions. After all, mammalian sperm flagella need to navigate through the intricate female reproductive tract[72], respiratory cilia beat in viscous mucous, and ependymal cilia beat in watery CSF. Detailed structure-function relationships, however, remain to be clarified in the future.

In summary, our multi-scale structural study revealed the high-resolution structure of the mammalian RS head-neck complex as well as the unique in situ axoneme structure of ependymal cilia. We propose a structural model for mammalian RS3, especially for its head, although further examination is needed. Our findings that ependymal cilia lack IDA-b/c/e, have simplified MIPs in DMT, and yet are sufficient to function in a water-like environment highlight an evolutionary choice driven by nature. Our flexibility and multi-scale structural analysis suggested the rigid, elastic, and rigid-elastic RS-CP contacts would constrain the relative movement between CP and RS, transfer the mechanical force between DMT and CP, and mediate the cilia mode of motion to collectively enable an asymmetric planar beat pattern of cilia. These findings could facilitate our understanding on the etiology of PCD/asthenospermia and on the dyskinesia of ependymal cilia in the development of hydrocephalus.

## Methods

### Plasmid construction

Genes encoding the full-length Rsph2 (NM_001360608) and Rsph16 (NM_153527) were amplified by PCR using total cDNAs from mouse ependymal cells (mEPCs), and were then constructed to pcDNA3.4 with N-terminal HA or N-terminal 3×FLAG tag, respectively, by Exonuclease III to express HA-tagged Rsph2 and FLAG-tagged Rsph16. Plasmids of Rsph1 (NM_001364916), Rsph3b (NM_001083945), Rsph4a (NM_001162957), Rsph9 (NM_029338), Rsph10b (XM_006504895) and Rsph23 (NM_080637) were previously cloned[27], in which Rsph1 and Rsph23 were subcloned into pcDNA3.4 with *Strep*-tag II and Myc tag at N-terminal, respectively. Non-tagged Rsph9 and Rsph3b were subcloned into pcDNA3.4, Rsph10b was subcloned into pcDNA3.1, and Rsph4a was subcloned into pMLink. To express GFP-tagged AK8, the full-length mouse AK8 was amplified by PCR using total cDNAs from mEPCs and subcloned into pLV-EGFP-C1 (Supplementary Fig. 1d). Primer sequences are available as Supplementary Data 1.

### Transfection and biochemical assays

The HEK293F cells (Gibco, Catalog Numbers A14527) were cultured in Dulbecco's modified Eagle's medium (DMEM) at 37 °C and a 5% $CO_2$ atmosphere, maintaining a density between 1 to $6 \times 10^6$ cells/mL. To prepare for transfection, the cells were diluted to a concentration of $1 \times 10^6$ cells/mL one day prior to transfection. The transfection mixture for 1 L of HEK293F cells were prepared by mixing a total of 1 mg (equally except Rsph2 and Rsph16 were doubled) of plasmids and 2.5 mg of polyethyleneimine (PEI) in 50 mL of DMEM medium separately, each was allowed to stand alone for 5 min. Subsequently, the plasmid solution and PEI solution were combined thoroughly and left to settle for 30 min before being added to the cell suspension. Cells were transfected by adding the mixture dropwise and were cultured at 37 °C and a 5% $CO_2$ atmosphere for 48 h; subsequently, harvested by centrifuged at $2000 \times g$ for 10 min and then suspended in 500 mL of PBS, pelleted again to remove the medium residue. Finally, the pellet was stored at −80 °C for further use.

## RS head-neck purification

The purification proceeded at 4 °C or in an ice bath, except for prompting. To purify the RS head-neck complex, cells from 2 L suspension were homogenized by cryo-milling and then dissolved in 200 mL of extraction buffer (25 mM HEPES pH 7.5, 200 mM KCl, 0.05% NP-40, 1 mM EDTA, 10 mM $Na_4P_2O_7$, 10% Glycerol, 1 mM DTT and protease inhibitor cocktail) at 4 °C by stirring. Subsequently, the dissolved suspension was filtered by 8 layers of medical gauze to remove big residue and then was centrifuged at 20,000 × g for 60 min to clarify the supernatant. The supernatant was incubated with balanced STarm Strep-tactin beads (Smart-Lifesciences, SA092100) overnight to allow bonding of the RS head-neck complex on beads. Then, the beads were washed by 200 mL of extraction buffer. To elute the complex, 10 mL of 50 mM D-biotin dissolved in extraction buffer was added and incubated with the Strep-tactin beads for 40 min. Afterwards, the beads were eluted by another 10 mL of 50 mM biotin buffer again. Eluates were pooled and incubated with balanced anti-FLAG beads (Smart-Lifesciences, SA042100) for 3 h. Subsequently, the resin was washed by 200 mL of extraction buffer and then eluted by 20 mL of 1 mg/mL 3 × Flag peptide dissolved in extraction buffer. To prepare cross-linked sample for cryo-EM, the eluate was cross linked through a 0–0.1% glutaraldehyde (GA) GraFix assay[73] in 10–40% glycerol at 4 °C for 15 h. The gradient was fractionated into 20 (each 500 μL) tubes and quenched by adding Tris-HCl (pH 7.5) to a final concentration of 50 mM. Fractions 16, 17, and 18 were used for cryo-EM grid preparation.

## Cryo-EM sample preparation and data collection

To prepare the cryo-EM sample of RS head-neck complex, freshly purified samples at a concentration of about 2 mg/mL were applied on a plasma-cleaned holy carbon grid (R1.2/1.3 or R2/1, Au, 300 mesh; Quantifoil). The grid was blotted using a Vitrobot Mark IV (Thermo Fisher Scientific) for 1 s at 100% humidity and 8 °C, and then plunged frozen into liquid ethane. Cryo-EM movies of the samples were collected on a Titan Krios electron microscope (Thermo Fisher Scientific) operated at an accelerating voltage of 300 kV. The movies were collected at a magnification of 81,000x and recorded on a K3 direct electron detector (Gatan) operated in the counting mode (yielding a pixel size of 0.854 Å), and under a low-dose condition in an automatic manner using EPU 2.11 software (Thermo Fisher Scientific). Each frame was exposed for 0.08 s, and the total exposure time was 2.41 s, leading to a total accumulated dose of 54 e$^-$/Å$^2$ on the specimen. For tilt movies, data were collected with stage tilt at 40° while other conditions remain unchanged.

## Cryo-EM image processing

A total of 29,585 non-tilt movies and 6,116 tilt movies were collected. MotionCor2[74] embedded in Relion 3.1[75,76] was used for motion correction, and CTFFind 4.1.8[77] was used for CTF estimation. Initially, 5.5 million particles were picked from the non-tilt movies using crYOLO 1.8.0[78], followed by extraction with a box size of 280 pixels in bin-by-2 mode. After reference-free 2D classification in cryoSPARC v4[79], ~3.5 million particles were kept. Ab-initio reconstruction in cryoSPARC based on 70,353 particles generated two initial models for subsequent 3D reconstruction, including a RS head-neck monomer and a dimer. After two rounds of heterogeneous refinement, 57% of the particles were classified into the monomer form and 43% into the dimer form. Similarly, ~1.7 million particles were picked from the tilt movies and then CTF estimation was performed by using goCTF 1.1.0[80]. After two rounds of heterogeneous refinement, 50% of the particles were classified into the dimer form and the other 50% into the monomer form. Unless otherwise described, the data processing was performed in cryoSPARC v4.

The monomer particles from both the non-tilt and tilt data were then combined together. The original monomer initial model together

with the ones low pass filtered to 20 Å, 30 Å, and 60 Å, respectively, were used as references for heterogeneous refinement. The best class consisting of 41% of the particle was reconstructed by nonuniform refinement, and the coordinates were then imported into Relion 3.1 for CTF refinement, and Bayesian polishing. The particles were then back imported into cryoSPARC for another round of heterogenous refinement. The two classes with good features especially robust density in the neck region were used for subsequent nonuniform refinement, which generated a consensus map of the RS head-neck monomer from 433,940 particles at a resolution of 3.28 Å. Local refinement on the relative stable core region was performed to further improve the map to the resolution of 3.14 Å. Local refinement on the neck and two arms with masks were performed respectively to improve the structure features of theses dynamic regions. In addition, in this dataset, a very minor population without the neck components was also detected and reconstructed to yield a RS head core map at the resolution of 7.93 Å.

For the head-neck dimer dataset, similar procedure was followed for the initial heterogeneous refinement. This yielded a good class consisting of 41% of the particles, which was then nonuniform refined without imposing any symmetry. The yielded dimmer map displayed a C2 symmetry between the two monomers. The particles were re-extracted and re-centered in Relion 3.1, followed by performing CTF refinement and Bayesian polishing. The particles were then back imported into cryoSPARC and cleaned up through 2D classification. 338,084 good particles were used for nonuniform refinement with imposed C2 symmetry, yielding a consensus RS head-neck dimer map at the resolution of 3.57 Å. Prior to local refinement, particles were subjected to symmetry expansion with C2 symmetry. The local refinement specifically targeting the Rsph16-Rsph16' bridge led to a resolution improvement to 3.36 Å. Furthermore, to improve the structure details of the monomer half within the dimer, the signal from the other half was subtracted from the symmetry-expanded dataset. Then, 3D classification was performed to remove bad particles, followed by a local refinement to yield a refined local monomer map at the resolution of 3.46 Å. On this basis, a further local refinement was performed on the neck region of the local monomer map, yielding an improved neck density at the resolution of 3.97 Å. Individual focus-refined maps were enhanced using DeepEMhancer 0.14[81]. To generate the composite maps for both the monomer and dimer, we aligned the focus-refined maps into the corresponding consensus map and merged them with the 'vop maximum' command in ChimeraX 1.5[82,83].

The overall resolutions for all of the cryo-EM maps in this study were determined based on the gold-standard criterion using a Fourier shell correlation (FSC) of 0.143. Moreover, we performed 3D flexible refinement on the dimer dataset in cryoSPARC v4 to capture its continuous conformational flexibility.

## Mice

All animal experiments were performed following guidelines approved by the Institutional Animal Care and Use Committee of CAS Center for Excellence in Molecular Cell Science, Institute of Biochemistry and Cell Biology, Chinese Academy of Sciences. Male or female wild-type C57BL/6 J mice of postnatal day 0 or 8-week-old were obtained from Shanghai SLAC Laboratory Animal for primary mouse ependymal cell culture or immunoprecipitation, respectively. These mice were accommodated under specific-pathogen-free (SPF) conditions in cages, with a light/dark cycle of 12/12 hours. The temperature of the room was kept between 22–24 °C with a relative humidity range of 45–65% humidity.

## EPC cilia preparation

Mouse EPCs were isolated and cultured as described[4]. Briefly, the telencephalon of postnatal day 0 mice was dissected with tweezers (Dumont, 1214Y84) in cold dissection solution (161 mM NaCl, 5 mM KCl, 1 mM $MgSO_4$, 3.7 mM $CaCl_2$, 5 mM HEPES, and 5.5 mM Glucose, pH

7.4) under a stereo microscope. The telencephalon was digested with 1 mL of the dissection solution containing 10 U/ml papain (Worthington, LS003126), 0.2 mg/mL L-Cysteine, 0.5 mM EDTA, 1 mM CaCl₂, 1.5 mM NaOH, and 0.15% DNase I (Sigma, D5025) for 30 min at 37 °C. Cells were dissociated by pipetting up and down 10 times with a 5 ml pipette and collected by centrifugation at $400 \times g$ for 5 min at room temperature. Cells were resuspended with DMEM (Invitrogen, 12430062) supplemented with 10% fetal bovine serum (FBS) and 50 µg/mL primocin (InvivoGen) and inoculated into the flask coated by 10 µg/mL fibronectin (Sigma, FC010). Neurons were shaken off and removed after culturing for 24 h and 48 h after inoculation. The remaining cells were further cultured to confluency, followed by culturing in serum-free DMEM medium supplemented with antibiotics for 10 days to induce multi-ciliation.

To better preserve the ultrastructure of ependymal cilia, we avoided using detergents and harsh conditions during cilia sample preparations. We detached multiciliated EPCs by trypsinization. Once cell detachment was achieved (1 to 2 min at 37 °C), serum-containing medium was added to neutralize trypsin. A centrifugation step at $350 \times g$ for 3 min at 4 °C was executed to remove the culture media. The cells were resuspended in PBS and centrifuged again. The majority of PBS was removed, leaving approximately 100 µL with the cell pellet. The EPCs were then resuspended by gently pipetting multiple times. The preparations, containing detached cilia and multiciliated cells, were used immediately for cryo-EM. In such a delicate procedure, a substantial number of cilia were still connected to cell bodies after vitrification, and ciliary membrane remained intact in most of the cilia (Supplementary Fig. S6).

### Cryo sample preparation and cryo-ET data collection

To prepare the cryo-ET sample of mouse ependymal cilia, the concentrated 10 nm colloidal gold fiducial solution (BBI Solution) was mixed with equal value of cilia, and a 3 µL aliquot of the sample was applied on a plasma-cleaned grid (R3.5/1 Cu 200 mesh; Quantifoil). The grid was blotted with Vitrobot Mark IV (Thermo Fisher Scientific) at 100% humidity and 8 °C, and then plunged into liquid ethane cooled by liquid nitrogen.

Cryo-ET movies of the samples were collected on a Titan Krios electron microscope (Thermo Fisher Scientific) operated at an accelerating voltage of 300 kV equipped with an energy filter (Gatan) operated at a slit width of 20 eV. Using SreialEM 3.8[84], the movies were collected at a magnification of 26,000× and recorded on a K3 direct electron detector (Gatan) operated in the counting mode (yielding a pixel size of 2.70 Å). Tilt-series were acquired from −60° to +60°, with 2° or 3° increments and two tilts per reversal using a dose-symmetric scheme[85]. The exposure time was 1 s or 1.5 s with 10 frames for each tilt angle, and the total dose was about 120 to 140 e⁻/Å². Tilt series were acquired at the targeted defocus setting of −3.0 - −5.0 µm.

### Cryo-ET data processing and sub-tomogram averaging procedure

All cryo-ET movie frames were corrected with a gain reference collected in the same EM session. Movement between frames was corrected using Warp 1.0.9[86] or MotionCor2[74] implemented in Relion 3.1[76]. Stacks with motion corrected averages were generated by Warp. Manual fiducial alignment and CTF-corrected tomogram reconstruction at bin4 were then performed in IMOD 4.11[87,88]. The bin4 tomograms were then CTF deconvolved with IsoNet 0.2.1[89] for better visualization and later particle picking.

After visual inspection of the originally collected 52 tilt series, 14 tomograms with undistorted "9 + 2" cilia ultrastructure and good contrast were chosen for further sub-tomogram averaging. 96-nm axonemal repeat with RS1/RS2/RS3 were manually picked using IMOD at the point where RS2 was rooted on the DMT, and 788 such 96-nm axonemal repeat were picked. Some sub-tomograms from good bin4 tomogram were used to generate an average in PEET (Particle Estimation for Electron Tomography) 1.15.1[9] as initial reference in Relion 4.0.

3D CTF estimation was performed in Warp using the alignment information from IMOD. The particle location, CTF estimation and stacks alignment information together with the motion corrected stacks were transferred to Relion 4.0 tomography pipeline for subsequent sub-tomogram averaging[90]. Pseudo-subtomos were made at bin4. These sub-tomograms were refined against the PEET generated initial reference, resulting in a 96-nm axonemal repeat with RS1/RS2/RS3. These particles were further re-extracted, refined, and 3D classified at bin2 (pixel size, 5.4 Å) and then bin1 (pixel size, 2.7 Å). This processing yielded a consensus sub-tomogram averaged map at 24.2 Å resolution from 606 sub-tomograms after frame alignment and CTF refinement.

To investigate dynamics and improve structural details of interested complexes such as RS1/RS2/RS3 and IDAs, focused classification was performed at bin4 or bin2. Take focused refinement on RS2 as an example, a good class consisting of 76.6% of the sub-tomograms were selected and refined at bin 2. Through further focused 3D classification on the RS2 head-neck, a major class (86.2%) exhibiting good feature was sorted out. Volume tracer function in UCSF Chimera 1.15[91] was used to determine the center of RS2 and RS2 head-neck, and then the RS2 and RS2 head-neck sub-volumes were, respectively, re-extracted and recentered at bin1. The recentered sub-volumes of RS2 and RS2 head-neck were refined to obtain a final map at the resolution of 24.2 Å and 23.8 Å, respectively.

To obtain a better DMT map with a periodicity of 48 nm, we divided the DMT portion of the 96-nm repeat map into two 48-nm repeat doublets, and re-extracted them to obtain 1,574 sub-volumes of the 48-nm repeat DMT. These sub-volumes were refined at bin2 and then bin1. After subsequent sub-volume CTF refinement and frame alignment, an in situ DMT map of mouse ependymal cilia was yielded at the resolution of 19.6 Å. All the refined maps were post-processed or filtered for visualization. The resolutions for maps were estimated based on FSC values of two independently refined half datasets (FSC = 0.143). IMOD was used to visualize the tomographic slices.

### Atomic model building

To build an atomic model for the RS head-neck complex, we used the atomic model of the mouse RS head core complex (PDB: 7DMP https://doi.org/10.2210/pdb7dmp/pdb) from our previous study[27] and AlphaFold2 predicted models for the neck subunits as initial model. We utilized the StarMap 1.2.15[92], a graphics-user-interface (GUI), to run Rosetta 2017 in ChimeraX[82,83]. We first performed pixel size calibration in Rosetta to yield a correct pixel size of 0.854 Å/pix, then fitted the initial model into the head-neck monomer map in ChimeraX by rigid body fitting and trimmed residues outside the density. Subsequently, we flexibly refined the model against the monomer composed map using Rosetta[93]. We then inspected and locally refined the model against the map in Coot 0.9.7[94]. Finally, we used the Real_space_refine module[95] in Phenix 1.19.2-4158[96] for the complete model refinement with the constraint of the map. For the atomic model of the RS head-neck dimer, as the map of the monomer fits in that of the dimer very well (Supplementary Fig. 3e), we fit two copies of the monomer model into dimer map and merged them together. Subsequently, we flexibly refined the model against the dimer composed map in Rosetta, and then in Phenix for real-space refinement of dimer.

### Co-immunoprecipitation

The plasmids expressing FLAG-Rsph3b and either GFP-AK8 or GFP were co-transfected into HEK293T cells for 48 h. Cells were lysed with a low-salt co-immunoprecipitation (CoIP) buffer (20 mM Tris-HCl pH 7.5, 100 mM KCl, 0.1% NP-40, 1 mM EDTA, 10% Glycerol, 10 mM Na₄P₂O₇, 1 mM DTT, PMSF and protease inhibitor cocktail) and cleared by

centrifugation at 14,500 x g for 20 min at 4 °C. The supernatants were incubated with 50 μL of anti-FLAG beads (Smart-Lifesciences, SA042005) for 3 h at 4 °C. The beads were washed three times with lysis buffer and three times with wash buffer (20 mM Tris-HCl pH 7.5, 150 mM KCl, 0.5% NP-40, 1 mM EDTA, 10 mM Na₄P₂O₇, 10% Glycerol). The proteins bound to the FLAG beads were eluted with 50 μL of 1 mg/ml FLAG peptide and subjected to western blotting analysis with anti-GFP (rabbit, MBL, Cat. # 598) and anti-FLAG (mouse, Sigma, Cat. # SLBN8915V) antibodies (Supplementary Fig. 7b).

To conjugate the antibodies, including rabbit IgG (ThermoFisher, 10500 C), anti-Rsph3 (Proteintech, 17603-1-AP), and anti-Ak8 (Merck, Hpa021445), to Protein A magnetic beads (Invitrogen, 10001D), an 8-hour incubation at 4 °C was carried out in a PBS buffer containing PMSF and a protease inhibitor cocktail. For the preparation of mouse testis lysates, the testis of 8-week-old mice was homogenized in a low-salt coIP buffer and then clarified by centrifugation at 6,000 x g for 20 min at 4 °C. The supernatant was subsequently incubated overnight at 4 °C with the antibodies conjugated to protein A magnetic beads. Following this, the beads were washed three times with wash buffer. The proteins were eluted using 0.1 M Glycine-HCl (pH 2.5) and the eluates were subjected to Mass Spectrometry analysis.

## Lentiviral production and infection
Lentiviral production and infection were carried out following previously described methods[97]. Briefly, HEK293T cells in 10 cm dishes were co-transfected with pLV-EGFP-AK8, pCMVdr8.9, and pMD2.VSVG for 48 h. The medium containing the virus was collected after centrifugation at 350 × g for 10 min to remove cell debris. Cultured mEPCs were infected one day before serum starvation (day -1), unless stated otherwise.

## Colocalization of AK8 and Rshp4a in cilia by super-resolution microscopy
After serum starvation for ten days, mEPCs cultured on 29 mm glass-bottom dish (Cellvis, D29-14-1.5-N) were fixed with 4% paraformaldehyde in PBS for 15 min at room temperature, followed by permeabilization with 0.5% Triton X-100 in PBS for 15 min. After blocking with 4% BSA in TBST for 1 h at room temperature, the cells were incubated with primary antibodies, mouse anti-acetylated tubulin (1:1000, Sigma-Aldrich, T6793), and rabbit anti-Rsph4a (1:500), diluted in the blocking buffer, overnight at 4 °C. Secondary antibodies used: goat-anti-rabbit Alexa Fluor546 (1:1000, Life Technologies, A-11035), goat-anti-mouse Alexa Fluor647 (1:1000, Life Technologies, A-31571). 2D-SIM images were captured using HiS-SIM (computational SR) equipped with a Plan Apo ×100/1.5 NA oil-immersion objective lens (Olympus). Serial Z-stack sectioning was performed at 100-nm intervals.

## Cross-linking mass spectrometry
The RS head-neck complex, purified from fraction No.17 of glycerol gradient ultra-centrifugation, was cross-linked using 5 mM BS3 (bis(sulfosuccinimidyl) suberate) from Thermo Fisher in an ice batch for 2 hrs. To halt the cross-linking reaction, Tris-HCl (pH 7.5) was added to a final concentration of 50 mM. The cross-linked proteins were precipitated with acetone, dried, and subsequently dissolved in 8 M urea, 100 mM Tris-HCl (pH 8.5). Reduction and alkylation were carried out by adding 5 mM TCEP (Thermo Scientific) and 10 mM Iodoacetamide (Sigma), respectively, and incubating at room temperature for 30 min. The protein mixture was then diluted four times and digested overnight with Trypsin at a ratio of 1:50 (w/w) (Promega). The resulting digested peptide solutions were desalted using a MonoSpinTM C18 column (GL Science, Tokyo, Japan) and dried. The peptide mixture was analyzed using a homemade 30 cm-long pulled-tip analytical column (75 μm ID packed with ReproSil-Pur C18-AQ 1.9 μm resin, Dr. Maisch GmbH) coupled with an Easy-nLC 1200 nano HPLC (Thermo Scientific, San Jose, CA) for mass spectrometry analysis.

Data-dependent tandem mass spectrometry (MS/MS) analysis was performed using an Orbitrap Eclipse Tribrid mass spectrometer with FAIMS Pro interface (Thermo Scientific, San Jose, CA). FAIMS alternated between CVs of −45V and −65V with a cycle time of 3 seconds. MS1 spectra were acquired at 60,000 resolutions, scanning from 300–1800 m/z, with 50% RF Lens and a maximum injection time of 50 ms. Precursors were filtered using monoisotopic peak determination set to peptide, charge state 3 to 8, dynamic exclusion of 30 seconds with ±10 ppm tolerance excluding isotopes. Precursors were isolated in the quadrupole with a 1.6 m/z window for MS2. Ions were collected for a maximum injection time of 70 ms, fragmented with stepped normalized HCD collision energies of 20, 25, and 30, and MS2 spectra were acquired in the orbitrap at 30,000 and 15,000 resolution with the first mass set to 110 m/z. MS scan functions and LC solvent gradients were controlled by the Xcalibur data system (Thermo Scientific). Cross-linked peptide pairs were identified using pLink2 software[98].

## Reporting summary
Further information on research design is available in the Nature Portfolio Reporting Summary linked to this article.

## Data availability
All data needed to evaluate the conclusions in the paper are present in the paper and/or the Supplementary Materials. Cryo-EM maps determined for the RS head-neck complex in monomer and dimer forms have been deposited at the Electron Microscopy Data Bank with accession numbers of EMD-37949 (monomer composite map), EMD-38004 (monomer consensus map), EMD-38028 (monomer core), EMD-38029 (monomer arm 1), EMD-38030 (monomer arm 2), EMD-38031 (monomer neck) and EMD-38020 (dimer composite map), EMD-38003 (dimer consensus map), EMD-38013 (Rsph16-Rsph16' bridge of dimer), EMD-38014 (half of dimer), EMD-38019 (neck of dimer), and the associated atomic models have been deposited in the Protein Data Bank with accession numbers of 8WZB (monomer) and 8X2U (dimer). The mouse ependymal cilia cryo-ET maps generated in this study have been deposited in EMDB under the following accession numbers: EMD-37104 (96-nm repeat DMT), EMD-37111 (48-nm repeat DMT), EMD-37114 (RS1), EMD-37116 (RS1 head), EMD-37117 (RS2), EMD-37118 (RS2 head), EMD-37119 (RS3), EMD-37120 (RS3 head). Source data are provided with this paper.

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

## Acknowledgements

We are grateful to the staffs of the NCPSS Electron Microscopy facility, Database and Computing facility, Mass Spectrometry facility, and Protein Expression and Purification facility for instrument support and technical assistance, and to Shanshan Wang from the Core Facility of Center for Excellence in Molecular Plant Sciences, CAS, for the ID mass spectrometric analysis. This work was supported by grants from the Strategic Priority Research Program of CAS (XDB37040103), the NSFC (32130056, 31991192 to X.Z., 31872714, 32270725), National Key R&D Program of China (2017YFA0503503 to Y.C., X.Z., and X.Y.), Shanghai Academic Research Leader (20XD1404200), Shanghai Pilot Program for Basic Research from CAS (JCYJ-SHFY-2022-008). X.C. was supported by China National Postdoctoral Program for Innovative Talents (BX2021310), the fellowship of China Postdoctoral Science Foundation (2022M713141), Shanghai "Super Postdoctoral" Incentive Plan, and Sanofi Scholarship Program.

## Author contributions

Y.C., X.Y. and X.Z. designed the experiments. X.M. expressed and purified the RS head-neck complex with involvement of J.Luo. and C.L. X.M. performed cryo-EM data analysis with the assistance of C.X. X.M. and Q.H. performed model buildings. C.P. performed the XL-MS experiments. B.Q. cultured mEPCs with involvement of Y.F. J.Li. and X.M. collected the cryo-ET data with assistance of R.M. and X.S. C.X. performed cryo-ET and sub-tomogram averaging with involvement of X.M. and Y.T. X.M. and J.Li. performed the model fitting for cryo-ET maps. C.F. recorded the fluorescence image of mEPCs. B.Q. recorded the mEPC cilia beating movie. X.M, C.X. and J. Li analyzed the cryo-EM data. Y.C., X.M, X.Y., X.Z., and C.X. wrote the manuscript with the input from J. Li, B.Q., J.Luo. and Q.H.

## Competing interests

The authors declare no competing interests.
