## [Peer Review File · Nature Communications]

Multi-scale structures of the mammalian radial spoke and divergence of axonemal complexes in ependymal ciliaREVIEWER COMMENTS

Reviewer #1 (Remarks to the Author):

In motile cilia, beating organelle in eukaryotic cells, the radial spoke (RS) is believed to be the transducer to propagate signals from the central pair (CP) to the peripheral doublet microtubules (DMT) and dynein motor proteins decorating DMT.

Atomic structure of the whole RS were solved from isolated *Chlamydomonas* RS (Brown group: Gui et al. 2021), while the atomic structures of the part of the human and *Chlamydomonas* RS heads were solved from expression, purification and in vitro reconstitution of component proteins, by Zheng et al. 2021 (the authors of this manuscript) and by Grossman-Haham et al. 2021 (Vale group), respectively. However, since lower-resolution studies of cilia by cryo-electron tomography demonstrated difference of RS between difference species, high resolution structural comparison between species was awaited.

In this work, Meng and colleagues solved atomic structure of the head-neck complex of mouse RS, consisting of eight component proteins (Rsph1,3b,4a,9,10b,2,23,16), by single particle cryo-EM and fit it to their new cryo-ET map from mouse ependymal cilia and modeled other RS components using AlphaFold2 and other modeling tools to discuss how RS components are networked in the physiological environment. Their RS head/neck preparation forms dimeric complexes, likely presenting heads/necks of RS1 and RS2, providing insight into the dimerization mechanism of RS1 and 2. They also discuss the art of interaction between the RS head and CP, which is likely the key of transducer mechanism.

The detailed description of protein connection in the mammalian RS in this work is novel. Fitting experimentally solved head/neck proteins and modeling of other components to fit to cryo-ET structure are convincing and present picture of how the mammalian RS is formed. This manuscript deserves publication in Nature Communications, after addressing the following points.

Major points:

The sentence 'IDA-b,c,e were all missing in ependymal cilia' in the chapter 'Reduced IDAs and MIPs in DMT of mouse ependymal axoneme': This type of conclusion must be carefully stated. It cannot be excluded that IDA-b,c,e are lost during cell separation (trypsin treatment), mechanical cilia isolation, which is harsh, (it is not clear if they demembrane by detergent) and cryo-EM preparation. Proteomic analysis of the whole cell with minimum stress by cell separation from membrane will help to examine IDA-b,c,e in ependymal cilia.

Their conclusion about lost Tektin (p.10, line6-7) also should be examined by MS.

Fig.3D: Fitting AK8 to the unassigned density is not convincing enough to identify this density as AK8. Is there any biochemical or proteomic proof to assign AK8 to this density?

p.9, line28: How do the subtomograms with and without IDA-a distribute in cilia? Do they have any DMT specificity, localization between distal and proximal parts or distribute randomly?

Minor points:

Title: The 'high resolution structure' in this work is especially about the head-neck complex of RS. This reviewer would advise the authors to take this in the title.

Citation in Introduction in the paragraph 'During the past years': It is not clear which reference is cited for which fact. These references should be mentioned item-by-item, such as 'including protists such as *Chlamydomonas*^{7,8,19,20} *Tetrahymena*⁷, *Choanoflagellate*¹⁸, and *Trypanosoma*, as well as metazoa such as human¹⁶ or bovine respiratory cilia and sea urchin, mouse^{21,22}, or human sperm flagella²¹'.

Fig.1: Presentation angles of the RS head/neck complex seem different from the one in their PNAS paper (ref.23), which makes comparison difficult. For this reviewer, the structures of the head/neck complex seems slightly different between this manuscript and the PNAS paper, but not for sure. It would be helpful to present the structure with the same view angle as Fig.2 of Zheng et al. (2021) PNAS as a supplementary figure and make a detailed description of their differences (or no difference).

End of the paragraph 'The RS head-neck complex exists as ...': The interpretation 'Rsph10b may not be a standard component of RS1 and RS2's head-neck complex.' is not clear. Do the authors think of any other protein, which replaces Rsph10b? Or do majority of RS lack proteins in the position of Rsph10b?

End of the paragraph 'Subunit interaction network of the RS...': 'Rsph16 also use the C-terminus' should read 'Rsph16 also uses'.

Line1 of p.8 and Fig.3D: 'Several Rsph22' is not clear. There cannot be several Rsph22s in one RS. Does it mean the Rsph22 of several RSs by classification?

Line24 of p.10: Comparison between their DRC structure and previously published *Chlamydomonas* DRC structure by Brown's group will be interesting.

Bottom of p.13: 'one of the DMTs bears lost IDA-c in *Chlamydomonas*' should read '...bears lost IDA-b in *Chlamydomonas*', according to Bui et al. 2012 JCB.

Reviewer #2 (Remarks to the Author):

"High-resolution structure of mouse radial spoke and its in-situ structure in ependymal cilia revealed by cryo-EM and cryo-ET" by Ming et al describes an investigation starting with revealing the structure of radial spoke (RS) subcomplex key to mechano-transduction, underlying rhythmic beating of cilia. The study employed powerful approaches such as expression of mammalian subunits in a cell line, followed by reconstitution, affinity purification, velocity centrifugation and newer imaging approaches – cryo-EM and cryo-ET of primary cell cultures - in conjunction with computation modeling. A similar strategy was taken by the team's recent publication (Zheng et al., 2021), with fewer subunits and a limited scope.

Reconstitution of a multitude of proteins into particles resembling the expected complicated configuration is by itself laudable and makes possible for high-resolution imaging. The averaging-aided imaging approaches - cryo-EM as well as cryo-ET (tomogram) - resolved RS structures of reconstituted particles and in the context of intact ependymal cilia, providing sufficient resolution for fitting atomic models with in vivo relevance. With the models, it is possible to position individual subunits and the key amino acids causative to the congenital disorders of dyskinetic cilia, offering molecular explanations for patients in the past and most likely in the future. Also discovered unexpectedly are the structures largely missing in the cilia in cultured ependymal cells, compared to motile cilia and flagella in other mammalian tissues and model organisms. In this reviewer's opinion, these structures ependymal cilia lack, not have including RS, is the most thought provoking and could be of interest to a broad audience. However, the writing, data presentations and discussion should be more precise and thoughtful in order to be convincing and to convey the key contributions clearly.

Title:

Worth of considering is an alternative title that are less about technicalities but more about the findings that many readers of this journal will appreciate, like "Divergence of high-resolution cryo-structures of the radial spoke and related complexes in ependymal cilia". The nuance of cryo-EM versus cryo-ET and purified particles versus in-situ super-complexes can be described elsewhere in detail for structural biologists to eschew. Mouse is not the key, whereas ependymal cilia already imply mammalian relevance and are a neglected area that has much to offer.

In Abstract,

"However, the high-resolution structure of mammalian RS remains missing" is unnecessarily provocative. The degree of resolution has evolved over the years since the advent of EM. The "high-resolution of mammalian RS" and diseases relevance were reported (for example, Lin et al., 2014; Zhao et al., 2021), although the resolution achieved then, as expected, was lower and the scope was narrower than shown in this study, partly empowered by the more recent introduction of computation analysis. What readers may likely appreciate is the atomic level of resolution in the structural differences reported here and the scientific insights in function and evolution.

The sentence "This tissue-specific feature may represent an evolutionary choice driven by the functional requirements on ependymal cilia." should be revised. It implies that ependymal cilia need to meet special requirements unique to circulate watery CSF. This is contrary to the less physical demands compared to that for mammalian sperm flagella and the other mammalian cilia in the lung or reproductive track which need to propel mucus with a far higher viscosity than the ventricular fluid, as stated later in this report.

" Our findings reveal the stepwise mammalian RS assembly mechanism, shed light on the coordinated rigid and elastic RS-CP interaction modes beneficial for the regulation of asymmetric ciliary beating, and also facilitate understandings on the etiology of ciliary dyskinesia-related ciliopathies and on the

ependymal cilia in the development of hydrocephalus.” “Reveal” is an overstatement since this structural study does not provide any evidence about assembly process. “shed light on” could be the common verb for the three major contributions proposed by the authors. Or based on the structure a stepwise assembly model could be proposed.

Introduction

P.3

“...motile cilia reside on the epithelial surface of various tissues, including the ependyma, trachea, and fallopian tubes...” is inappropriate. Trachea and fallopian tubes are not tissues.

“organ inversions” could be mis-interpreted and inversion is only one of situs anomalies caused by ciliary dyskinesia. “organ misplacement” may be more appropriate.

The paragraph “As CSF is rich in neuropeptides, its orderly flow is critical for nourishing the central nervous system and maintaining proper body axis. Dyskinesia of ependymal multicilia leads to the obstruction of CSF flow, resulting in hydrocephalus and idiopathic scoliosis³⁻⁵.” needs attention. Multi in multicilia is unnecessary. Cilia is a plural term. The two sentences, while likely correct individually to some degrees, appear disjoint, illogical together. The first is about ependymal cilia in distributing chemical nutrients, whereas the second talks about ciliary defects will result in physical blockage, presumably increasing ventricular pressure and thus hydrocephaly. Finally, the statement may not be entirely applicable to humans. While hydrocephaly is common in rodents and canines with ependymal cilia dyskinesia, the impact of dyskinesia cilia is often not evident or definitive in humans with such congenital anomalies.

Results:

P.4

The definitions for RS head in Zheng et al., 2021 and RS head, head-neck complex, RS monomer and RS dimer are imprecise. In the previous study, the RS head contains only Rsph1, 4a, 9 and 3b. In the current study, 4 more components not entirely restricted to the neck were included. As such the text and the proposed stepwise assembly model includes part of the new components in the RS head. The inconsistency is confusing but can be resolved by simply referring the reconstituted particle in the previous study is a partial RS head, whereas RS monomer should be a head-neck monomer. Likewise, RS dimer should be a head-neck dimer. There are additional components in the RS, regardless of monomer and dimer.

Figure 1 can be presented in a more persuasive way, to justify the corresponding text and its legend - Cryo-EM structures of mouse RS head-neck complex. Firstly, the legend should state this is reconstituted,

ie not naturally isolated, complex. The omission could lead readers to presume they are bona fide RS particles harvested from ciliated cells. Secondly, contrary to the legend, the figure only shows anatomic models, while a cryo-EM montage is relegated to the supplemental data (S1 and S2). Both data sets, preferentially including the overlaid atomic models and representative cryo-EM, are important for readers to assess the claimed 3-4 Å resolution and the fit independently. Furthermore, the recent studies of protist RS (Gui et al., 2021; Grossman-Haham et al., 2021) showed more polypeptides in this region than previously recognized. The original data will allow readers to determine how similar the cryo-EM of the reconstituted particles of only recognized RS proteins resemble the published cryo-EM of RS from various organisms and organs and whether mammalian RS might also have additional components. Modeling should not replace original data.

The claimed resolution in p. 4 “Our cryo-EM analysis on the assembled mouse RS head-neck complex revealed the presence of both monomer and dimer (Fig. S1C-D). We determined their structures at a resolution of 3.28 Å and 3.57 Å, respectively, with the head core region focus-refined to a resolution of 3.14 Å (Fig. 1B-C, S2, S3A-C, Table S1).” seems inconsistent with Fig S1C-D, which is consistent with the sentence in p. 6 “Our local resolution analysis of the RS head-neck dimer revealed relatively lower local resolution at the very end of the Rsph1 arm and the neck (Fig. S3B)”. Are the stated resolution in both pages inconsistent? Are the stated resolution applicable to Fig. S1C-D?

P. 5

Abbreviations, like XL-MS, should be spelled out when referred the first time. The crosslinker and the properties, at least including the nature of crosslinking and the arm length, should be included for readers to determine independently the validity of proposed molecular proximity and whether crosslinking data “supports” the modeling and the fit as narrated.

In “wrapped around by the previously missing N-terminal Dpy-30 motif of the head

subunits Rsph4a and Rsph4a’ (Fig. 1D)”, the underline space may benefit from the word “degenerate” and the reference. The DPY-30 motif in RSP4 was revealed by structural studies (Gui et al., 2021) and cannot be found by mere protein sequence analysis. Fig. 1D is not helpful in this regard.

“Unlike the extensive interplays among the other subunits, however, Rsph10b only exhibits limited association with Rsph16 in the XL-MS results (Fig. 1F), suggesting that Rsph10b may not be a standard component of RS1 and RS2’s head-neck complex.” This sentence is confusing. It assumes that extensive interplays and/or extensive association with Rsph16 is a feature of core head-neck proteins. Also, core may be a better word than standard.

The narrations before and after the section title “Subunit interaction network of the RS head-neck complex” in p. 5, seem to contain redundant descriptions about the key proteins and referral to the

preceding publication. Reorganization may lessen the redundancy. In addition, it is unclear whether most descriptions in this section about conserved core proteins are any different from that in algal counterpart (Gui et al., 2021). It is good to have a summary statement and highlight the differences now, rather than later.

P. 7

The conclusion of the middle paragraph, “Collectively, these data suggested that our recombinant RS head-neck complex represents the endogenous head-neck architecture of both RS1 and RS2, thus further confirming that the mammalian RS head complex is distinct in composition from the *Chlamydomonas* one^{7,23-25,27}.” is perplexing. The underlined phrase conflicts with the established facts and previous statements naming human spoke components after algal counterparts founded on extensive homologies in structural motifs and sequence similarities among orthologs throughout evolution. Wording with precision in this case is necessary to reflect the facts and does not diminish the contribution of this study.

P. 8

In “...we then fitted Cfap61 and Cfap251, the homolog of Fap61 and Fap251”, it is not clear at all if C is indicative of *Tetrahymena*, *Chlamydomonas*, Human or Mice. The paragraph in current state does not make sense.

P. 9

In “two RS1 conformations relative to the DMT”, conformation, meaning form or shape, is inappropriate for the slight tilt of the complex.

In “... RS1 may be more dynamic in regulating cilia motion...”, it is unclear whether the authors imply RS1 is inherently more dynamic like changing position or conformation, or merely structurally more flexible, or is more susceptible to induced tilt upon interacting with the central pair apparatus, as shown in another ciliated organism (Warner and Satir, 1974).

“In contrast to the helical beat form of sperm flagella, mammalian ependymal and respiratory multicilia beat in a back-and-forth manner but differ greatly in number.....”. This sentence is so simplified that it becomes inaccurate. Sperm flagella exhibit a variety of waveform, depending on the species, the activation state, and the environments, for example. The term helical beat form, which implies 3-D, does not do the justice. If ciliary motility parameters are not the focus of this study, it suffices to state simply differences in waveform, number.....

P. 10

“...the mouse sperm flagella exhibit much more filamentous MIPs that is similar to but more extensive than that of tektin filaments inside the A-tubule of DMT (Fig. 4C)22.” are?

In the N-DRC paragraph, it is unclear what the main point from the exhaustive structural descriptions is. What do they mean? Is it same as or different from that in other systems and organisms and systems? The detail structures of the axoneme cannot fit into one paper. This section could go to supplemental data, if it does not add to the coherent story.

P. 11

In “.... collectively indicating a potentially more complex RS-CP interaction mode in mammals.” it is not evident from the sentence why the authors think the different chemical forces in RS-CO interactions means more complex in mammalian modes.

Discussion

It seems that in p. 11 “...intrinsic modes of motion of this regulatory machinery of ciliary motility (Fig. 2A, C).” and “the intrinsic dynamics of the head-neck dimer” in the Fig. 2 legend convey the same concept. Intrinsic dynamics seems to be different from intrinsic flexibility and imply inherent mobility. Albeit possible, this cannot be demonstrated by solid phase imaging in this study. Additional modest interpretations should be included, like the structural variations imply flexibility which may allow RS transduce forces between the central pair and DMTs as cilia in natural environments bend or twist, not limited to rhythmic beating with identical waveform and beat frequency.

In “.....sperm-specific component densities cross-linking RS1/RS2/RS3 were absent...”, alternative terms - like connecting or linking - may be more suitable than cross-linking, given cross-linking is used in XL-MS.

“Collectively, our study sheds new lights on the stepwise assembly mechanism of mammalian RS1/RS2....” overstated. The authors certain earn the right to propose a stepwise assembly model. Experiments are needed in this case to shed the light.

P. 12

“Noteworthy, RspH16, the homologue of HSP40 and also comprising a C-terminal dimerization domain35, appears to play an essential role in the dimerization of the RS head-neck complex, which is key for the formation and stabilization of the RS1/RS2.” The structural evidence in this study did not

provide evidence in assembly, ie the formation. In fact, the interpreted essential role in the formation is not supported by the genetic evidence. Contrary to the prediction, RS1 and RS2 in algal mutants of RSP16 display no evident defects in compositions of RS1 and RS2, other than sporadic aberrant morphology and tilts.

P. 13

“... ependymal cilia may not require tektins within the A-tubule to provide extra stabilization/stiffness of DMT against waveform-dependent mechanical forces.” is awkward. The preceding statements are about less viscous environment, not about waveform. A revision is needed to make the argument sensible.

In “the IDAs which generate the waveform of...” perhaps determine is more appropriate.

For “Although it has been reported that IDA-b/e are missing from at least two DMTs in mouse respiratory cilia⁶⁵, and one of the DMTs bears lost IDA-c in *Chlamydomonas*, the complete loss of IDA-b/c/e in all DMTs has not been observed in any other tissues.”, the underlined statement needs a reference and revision. “bears” does not make sense.

P. 14

The middle paragraph is illogical in the current state. The authors seem to use the different amounts of MIPs and IDA species to explain the differences in the environment and required function of mammalian sperm flagella that need to navigate tortuous female reproductive track, cilia in viscous environments and ependymal cilia in CSF of low viscosity. But refinements are needed to make the argument understandable.

The authors should discuss the possibility that the missing key structures in cilia of cultured ependymal cells may only happen in culture.

Materials and Methods

P. 18

Reference is needed for “Mouse EPCs were isolated and cultured as described.”

REVIEWER COMMENTS

Reviewer #1 (Remarks to the Author):

In motile cilia, beating organelle in eukaryotic cells, the radial spoke (RS) is believed to be the transducer to propagate signals from the central pair (CP) to the peripheral doublet microtubules (DMT) and dynein motor proteins decorating DMT.

Atomic structure of the whole RS were solved from isolated Chlamydomonas RS (Brown group: Gui et al. 2021), while the atomic structures of the part of the human and Chlamydomonas RS heads were solved from expression, purification and in vitro reconstitution of component proteins, by Zheng et al. 2021 (the authors of this manuscript) and by Grossman-Haham et al. 2021 (Vale group), respectively. However, since lower-resolution studies of cilia by cryo-electron tomography demonstrated difference of RS between different species, high resolution structural comparison between species was awaited.

In this work, Meng and colleagues solved atomic structure of the head-neck complex of mouse RS, consisting of eight component proteins (Rsph1,3b,4a,9,10b,2,23,16), by single particle cryo-EM and fit it to their new cryo-ET map from mouse ependymal cilia and modeled other RS components using AlphaFold2 and other modeling tools to discuss how RS components are networked in the physiological environment. Their RS head/neck preparation forms dimeric complexes, likely presenting heads/necks of RS1 and RS2, providing insight into the dimerization mechanism of RS1 and 2. They also discuss the art of interaction between the RS head and CP, which is likely the key of transducer mechanism.

The detailed description of protein connection in the mammalian RS in this work is novel. Fitting experimentally solved head/neck proteins and modeling of other components to fit to cryo-ET structure are convincing and present picture of how the mammalian RS is formed. This manuscript deserves publication in Nature Communications, after addressing the following points.

--We appreciate the encouraging comments and insightful suggestions from this reviewer.

Major points:

Q1-1. The sentence 'IDA-b,c,e were all missing in ependymal cilia' in the chapter 'Reduced IDAs and MIPs in DMT of mouse ependymal axoneme': This type of conclusion must be carefully stated. It cannot be excluded that IDA-b,c,e are lost during cell separation (trypsin treatment), mechanical cilia isolation, which is harsh, (it is not clear if they demembrane by detergent) and cryo-EM preparation. Proteomic analysis of the whole cell with minimum stress by cell separation from membrane will help to examine IDA-b,c,e in ependymal cilia.

A1-1: Thanks for the insightful suggestion from our reviewer. In early pilot experiments, we used purified cilia detached from EPCs through mechanical shaking in a solution containing Triton X-100 and indeed observed an impaired ultrastructural integrity. We tested different procedures and finally chose the one involving only mild trypsinization to detach mEPCs

from culture dishes and gentle pipetting in PBS (without detergent) to partly detach cilia. In such a delicate procedure, a substantial number of cilia were still connected to cell bodies after vitrification (Fig. R1a-b), and ciliary membrane (arrowheads) remained intact in most of the cilia (Fig. R1c). We have presented these results in Supplementary Fig. 6 in our revised manuscript.

We apologize for not providing detailed cilia sample handling process in our manuscript. For better clarity, we have included details in the Methods section in the revised manuscript: “To better preserve the ultrastructure of ependymal cilia, we avoided using detergents and harsh conditions during cilia sample preparations. We detached multiciliated EPCs by trypsinization. Once cell detachment was achieved (1 to 2 min at 37°C), serum-containing medium was added to neutralize trypsin. A centrifugation step at 350 × g for 3 min at 4°C was executed to remove the culture media. The cells were resuspended in PBS and centrifuged again. The majority of PBS was removed, leaving approximately 100 μL with the cell pellet. The EPCs were then resuspended by gently pipetting multiple times. The preparations, containing detached cilia and multiciliated cells, were used immediately for cryo-EM. In such a delicate procedure, a substantial number of cilia were still connected to cell bodies after vitrification, and ciliary membrane remained intact in most of the cilia (Supplementary Fig. S6).” (L. 573-583 on P. 19-20).

Fig. R1 Quality of cilia preparations for cryo-ET imaging. (a) A representative search-mode image showing a bundle of cilia (yellow rectangle, bottom-left) connected to cell body (red circle). (b) A representative enlarged view showing four cilia connected to a cell body. (c) Four representative tomograms showing cilia with intact membrane (indicated by red arrowheads).

Following the request of our reviewer, we performed label-free quantitative mass spectrometry of whole EPCs by directly lysing the cells in the dish. In the MS data, representative RS head-neck components (Rsph1, -3b, -4a, -9) were detected at high abundance, with label-free quantification (LFQ) intensities ranging from 1.7k to 6k (Table R1). These subunits can serve as reference components. The MS hit 10 Dnah proteins (Table R1). In contrast to the high LFQ intensities (>439) of other Dnahs, including Dnah1 (IDA-d), Dnah2 (IDA-l1/f β), Dnah6 (IDA-g), and Dnah10 (IDA-l1/f α), the LFQ intensities of Dnah7c (~119) and Dnah12 (~103) were both low (Table R1). The low abundance of Dnah12 is consistent with our structural results that IDA-a is absent in some repeats (Fig. 4e). As Dnah7 has been assigned to both IDA-b and IDA-e in a very recent publication¹, the low abundance of Dnah7c is also consistent with the absence of IDA-b and IDA-e in our structure (Fig. 4a, b), though it remains to be clarified in the future whether the expressed Dnah7c is located in the cell body or in a portion of repeats that we missed. Dnah3, which has also recently been assigned to IDA-c in the same publication¹, however, was the most abundant of all Dnahs (~4,759; Table R1). As it is unlikely that our delicate sample preparations selectively resulted in the complete shedding of IDA-c from the axonemes (Fig. 4a, b), whether Dnah3 is correctly assigned to IDA-c remains to be an open question. Furthermore, it is also unclear to what extent a protein abundance detected in whole-cell MS can be used to refer to its abundance in the axonemes. Due to these complexities, we choose not to present the whole-cell MS results in our revised manuscript.

Since it is currently not possible to absolutely exclude that IDA-b,c,e are lost during the sample preparation, we have also included a statement in the Discussion section to indicate this possibility (“Alternatively, the absence of certain key structures might result from cell separation or cryo-vitrification due to the difficulty to absolutely exclude this possibility at present”). (L. 428-430 on P. 14-15).

Table R1 Representative mass spectrometry results of whole EPCs

Protein Groups	Genes	Protein Descriptions	LFQ intensity
Q3UFY4;Q9DA80	Rsph3a;Rsph3b	Radial spoke head protein 3 homolog A; Radial spoke head protein 3 homolog B	1774.108643
Q8BYM7	Rsph4a	Radial spoke head protein 4 homolog A	5820.134277
Q8VIG3	Rsph1	Radial spoke head 1 homolog	3105.580811
Q9D9V4	Rsph9	Radial spoke head protein 9 homolog	5983.522461
E9Q8T7	Dnah1	Dynein axonemal heavy chain 1	795.43573
P0C6F1	Dnah2	Dynein axonemal heavy chain 2	937.2059937
Q8BW94	Dnah3	Dynein axonemal heavy chain 3	4759.521973
Q8VHE6	Dnah5	Dynein axonemal heavy chain 5	2227.835449
E9Q0B6	Dnah6	Dynein, axonemal, heavy chain 6	1729.462891
A0A087WR13	Dnah7c	Dynein, axonemal, heavy chain 7C	119.2969742
B1AR51	Dnah9	Dynein, axonemal, heavy chain 9	3767.785889
D3YYQ8	Dnah10	Dynein, axonemal, heavy chain 10	439.4195862
E9Q7N9	Dnah11	Dynein, axonemal, heavy chain 11	1094.265015
Q3V0Q1	Dnah12	Dynein axonemal heavy chain 12	103.2020111
Q149S1	Tekt4	Tektin-4	252.0562134
Q9DAJ2	Tekt1	Tektin-1	471.074585

Q1-2. Their conclusion about lost Tektin (p.10, line6-7) also should be examined by MS.

A1-2: As described in A1-1, we performed proteomic analysis of whole EPCs. Tektin-2/3 and TEKTI1P1 were not detected in the label-free quantitative MS, and Tektin-1/4 were hit at relatively low abundance (LFQ intensity ≤ 471) (Table R1). As Tektins may also localize to basal bodies², these results generally support the observation of absent Tektin bundle within the A-tubule of DMT (Fig. 4c) but cannot be used as solid evidence (explained in A1-1). Furthermore, as explained in A1-1, the gentle process in sample preparations and intact ciliary membranes observed in tomograms (Fig. R1c) already minimized the possibility that the loss of MIPs was an artifact. We thus choose not to present the whole-cell MS results in our revised manuscript. Since it is currently not possible to absolutely exclude that Tektins are lost during the sample preparation, we have included a statement in the Discussion section to indicate this possibility (“Alternatively, the absence of certain key structures might result from cell separation or cryo-vitrification due to the difficulty to absolutely exclude this possibility at present”). (L. 428-430 on P. 14-15)

Q1-3. Fig.3D: Fitting AK8 to the unassigned density is not convincing enough to identify this density as AK8. Is there any biochemical or proteomic proof to assign AK8 to this density?

A1-3: Following the request, we have provided two additional lines of supporting evidence in the revised manuscript (Supplementary Fig. 7). Firstly, we expressed GFP-AK8 in cultured mEPCs and examined its subcellular localization. Super-resolution fluorescence microscopy confirmed that GFP-AK8 co-localizes with Rsph4a, an RS marker, in the axonemal central lumen of ependymal cilia (Supplementary Fig. 7a). Secondly, as our model (Fig. 3d) implicated an interaction between Rsph3b and AK8, we co-expressed FLAG-Rsph3b and GFP-AK8 in HEK293T cells and readily detected an association between the two proteins through co-immunoprecipitations (Supplementary Fig. 7b). We hope that our reviewer would agree that these data significantly strengthen the relationship between AK8 and the RS.

Q1-4: p.9, line28: How do the subtomogram with and without IDA-a distribute in cilia? Do they have any DMT specificity, localization between distal and proximal parts or distribute randomly?

A1-4: To address these questions, we performed *in situ* IDA-a distribution analysis and found that IDA-a tends to be either mostly present (Fig. R2a) or completely absent (Fig. R2b) in the tomograms. Furthermore, the missing IDA-a occurred in different DMTs (Fig. R2a), suggesting a lack of DMT specificity. Nevertheless, since we did not record the relative location of each tomogram within the cilia, we are unable to determine relative locations of individual subtomogram on the cilia, nor could we estimate their distribution patterns.

Fig. R2 *In situ* IDA-a distribution analysis. (a) A representative axonemal tomogram showing the presence of most IDA-a. (b) A representative tomogram showing the absence of all IDA-a. Magenta puncta represent IDA-a in axonemal repeats, whereas cyan puncta indicate missing IDA-a.

Minor points:

Q1-5. Title: The ‘high resolution structure’ in this work is especially about the head-neck complex of RS. This reviewer would advise the authors to take this in the title.

A1-5: We thank our reviewer for the comment and advice. As reviewer #2 also commented on the title, we have taken these suggestions and changed the title to “Multi-scale structures of the mammalian radial spoke and divergence of axonemal complexes in ependymal cilia” in the revised manuscript.

Q1-6. Citation in Introduction in the paragraph ‘During the past years’: It is not clear which reference is cited for which fact. These references should be mentioned item-by-item, such as ‘including protists such as Chlamydomonas^{7,8,19,20} Tetrahymena⁷, Choanoflagellate¹⁸, and Trypanosoma, as well as metazoa such as human¹⁶ or bovine respiratory cilia and sea urchin, mouse^{21,22}, or human sperm flagella²¹’.

A1-6: We have followed the suggestion and cited the references item-by-item (L. 73-76 on P. 3).

Q1-7. Fig.1: Presentation angles of the RS head/neck complex seem different from the one in their PNAS paper (ref.23), which makes comparison difficult. For this reviewer, the structures of the head/neck complex seem slightly different between this manuscript and the PNAS paper, but not for sure. It would be helpful to present the structure with the same view angle as Fig.2 of Zheng et al. (2021) PNAS as a supplementary figure and make a detailed description of their differences (or no difference).

A1-7: The reviewer is right about the presentation angles. In the revised manuscript, we have illustrated the complex as requested (Supplementary Fig. 3f) and additionally included an overlaid structural comparison between our previous RS head core³ and the

current RS head-neck monomer (Supplementary Fig. 3g). There is no structural difference in the head core portion between the two maps.

Q1-8: End of the paragraph 'The RS head-neck complex exists as ...': The interpretation 'Rsph10b may not be a standard component of RS1 and RS2's head-neck complex.' is not clear. Do the authors think of any other protein, which replaces Rsph10b? Or do majority of RS lack proteins in the position of Rsph10b?

A1-8: We apologize for the confusion. We have modified the sentences to improve the clarity in the revised manuscript: "Notably, although Rsph10b appeared to be moderately abundant in the affinity-purified reconstituted RS head-neck complex (Supplementary Fig. 1b), it was only weakly detected by XL-MS (Fig. 1i) and completely absent in the RS1/RS2 head-neck maps (Fig. 1c, f) after subjecting the sample to an additional round of glycerol gradient centrifugation. Such results suggest that Rsph10b and some of the head-neck subunits may form a complex distinct from the head-neck complex of RS1/RS2." (L. 137-142 on P. 5-6).

Q1-9: End of the paragraph 'Subunit interaction network of the RS...': 'Rsph16 also use the C-terminus' should read 'Rsph16 also uses'.

A1-9: Thanks for pointing this out. We have corrected this to "Rsph16 also uses" (L. 157 on P. 6).

Q1-10. Line1 of p.8 and Fig.3D: 'Several Rsph22' is not clear. There cannot be several Rsph22s in one RS. Does it mean the Rsph22 of several RSs by classification?

A1-10: We apologize for the confusion. It has been reported that the stalk and base regions of *Chlamydomonas* RS1 and RS2 respectively contain 8 and 12 copies of RSP22⁴. While we similarly incorporated 12 copies of Rsph22 into the mammalian RS2 (Fig. 3e), 8 copies of Rsph22 still failed to occupy the extra density of the RS1 (Fig. 3d). Based on the good fit between the model and the map as well as our MS analysis (Table S7), we placed one copy of AK8 in the stalk region of RS1 together with five copies of Rsph22 (Fig. 3d). To improve the clarity, we have accordingly modified the main text (L.232-245 on P. 8-9) and figure legend in the revised manuscript.

Q1-11. Line24 of p.10: Comparison between their DRC structure and previously published *Chlamydomonas* DRC structure by Brown's group will be interesting.

A1-11: We appreciate the suggestion. Since the *Chlamydomonas* N-DRC map by Brown's group is not released⁴, we compared our *in situ* N-DRC map of ependymal cilia with the *in situ* N-DRC maps in two earlier publications from Dr. Nicastro's lab on *Chlamydomonas* flagella⁵ (EMD-20338) and human respiratory cilia⁶ (EMD-5950), respectively (Fig. R3). Unlike the other two maps, the ependymal cilia N-DRC exhibits a reduced distal lobe density containing the C-terminal portions of DRC9/10, suggesting a weakened interaction between DRC9/10 and the neighboring DMT in ependymal cilia.

However, since the *in situ* N-DRC structure is not a main finding in this study, we followed the suggestion of reviewer #2 and have moved the contents in Fig. 4D into supplemental

data (now as Supplementary Fig. 8d in the revised manuscript). In the figure caption, we briefly mention that “the distal right lobe density is rather weak” (L. 895-898 on P. 45).

Fig. R3 Overlaid *in situ* N-DRC map of ependymal cilia (in green, this study) with *in situ* map of *Chlamydomonas* flagella⁵ (a, in transparent grey, EMD: 20338) or human respiratory cilia⁶ (c, in transparent blue, EMD: 5950). The reduced distal right lobe density in the ependymal cilia N-DRC is indicated by red dotted ellipsoid.

Q1-12: Bottom of p.13: ‘one of the DMTs bears lost IDA-c in *Chlamydomonas*’ should read ‘...bears lost IDA-b in *Chlamydomonas*’, according to Bui et al. 2012 JCB.

A1-12: The reviewer is right. We have corrected this mistake in the revised manuscript (L. 421 on P. 14).

Reviewer #2 (Remarks to the Author):

“High-resolution structure of mouse radial spoke and its in-situ structure in ependymal cilia revealed by cryo-EM and cryo-ET” by Ming et al describes an investigation starting with revealing the structure of radial spoke (RS) subcomplex key to mechano-transduction, underlying rhythmic beating of cilia. The study employed powerful approaches such as expression of mammalian subunits in a cell line, followed by reconstitution, affinity purification, velocity centrifugation and newer imaging approaches – cryo-EM and cryo-ET of primary cell cultures - in conjunction with computation modeling. A similar strategy was taken by the team’s recent publication (Zheng et al., 2021), with fewer subunits and a limited scope.

Reconstitution of a multitude of proteins into particles resembling the expected complicated configuration is by itself laudable and makes possible for high-resolution imaging. The averaging-aided imaging approaches - cryo-EM as well as cryo-ET (tomogram) - resolved RS structures of reconstituted particles and in the context of intact ependymal cilia, providing sufficient resolution for fitting atomic models with in vivo relevance. With the models, it is possible to position individual subunits and the key amino acids causative to the congenital disorders of dyskinetic cilia, offering molecular explanations for patients in the past and most likely in the future. Also discovered unexpectedly are the structures largely missing in the cilia in cultured ependymal cells, compared to motile cilia and flagella in other mammalian tissues and model organisms. In this reviewer’s opinion, these structures ependymal cilia lack, not have including RS, is the most thought provoking and could be of interest to a broad audience. However, the writing, data presentations and discussion should be more precise and thoughtful in order to be convincing and to convey the key contributions clearly.

--- We thank the reviewer for the comments and constructive suggestions. We hope our reviewer will find that the revised manuscript is significantly improved in the clarity of presentation.

Title:

Q2-1. Worth of considering is an alternative title that are less about technicalities but more about the findings that many readers of this journal will appreciate, like “Divergence of high-resolution cryo-structures of the radial spoke and related complexes in ependymal cilia”. The nuance of cryo-EM versus cryo-ET and purified particles versus in-situ super-complexes can be described elsewhere in detail for structural biologists to eschew. Mouse is not the key, whereas ependymal cilia already imply mammalian relevance and are a neglected area that has much to offer.

A2-1: We are grateful for the constructive suggestion. As reviewer #1 also commented on the title, we have taken these suggestions and changed the title to “Multi-scale structures of the mammalian radial spoke and divergence of axonemal complexes in ependymal cilia” in the revised manuscript.

In Abstract,

Q2-2. “However, the high-resolution structure of mammalian RS remains missing” is unnecessarily provocative. The degree of resolution has evolved over the years since the advent of EM. The “high-resolution of mammalian RS” and diseases relevance were reported (for example, Lin et al., 2014; Zhao et al.,2021), although the resolution achieved then, as expected, was lower and the scope was narrower than shown in this study, partly empowered by the more recent introduction of computation analysis. What readers may likely appreciate is the atomic level of resolution in the structural differences reported here and the scientific insights in function and evolution.

A2-2: We appreciate these comments and have modified the sentence as “Atomic-resolution structures of metazoan RS and structures of axonemal complexes in ependymal cilia, whose rhythmic beating drives the circulation of cerebrospinal fluid, however, remain obscure.” in the revised manuscript (L. 28-30 on P. 2).

Q2-3. The sentence “This tissue-specific feature may represent an evolutionary choice driven by the functional requirements on ependymal cilia.” should be revised. It implies that ependymal cilia need to meet special requirements unique to circulate watery CSF. This is contrary to the less s physical demands compared to that for mammalian sperm flagella and the other mammalian cilia in the lung or reproductive track which need to propel mucus with a far higher viscosity than the ventricular fluid, as stated later in this report.

A2-3: We appreciate the comments and have removed the speculation from the abstract. Now this part of the abstract reads as “Strikingly, our cryo-ET map reveals the lack of IDA-b/c/e and the absence of Tektin filaments within the A-tubule of doublet microtubules in ependymal cilia compared with mammalian respiratory cilia and sperm flagella, further exemplifying the structural diversity of mammalian motile cilia” (L. 35-38 on P. 2).

Q2-4. “Our findings reveal the stepwise mammalian RS assembly mechanism, shed light on the coordinated rigid and elastic RS-CP interaction modes beneficial for the regulation of asymmetric ciliary beating, and also facilitate understandings on the etiology of ciliary dyskinesia-related ciliopathies and on the ependymal cilia in the development of hydrocephalus.” “Reveal” is an overstatement since this structural study does not provide any evidence about assembly process. “shed light on” could be the common verb for the three major contributions proposed by the authors. Or based on the structure a stepwise assembly model could be proposed.

A2-4: As suggested, we have replaced “reveal” with “shed light on” in the revised manuscript (L. 38-39 on P. 2).

Introduction

Q2-5. P.3 “...motile cilia reside on the epithelial surface of various tissues, including the ependyma, trachea, and fallopian tubes....” is inappropriate. Trachea and fallopian tubes are not tissues.

A2-5: We have revised the description accordingly, to read: "...motile cilia reside on the surface of epithelial cells in the ependyma, trachea, and fallopian tubes, ..." (L. 46-48 on P. 3).

"organ inversions" could be mis-interpreted and inversion is only one of situs anomalies caused by ciliary dyskinesia. "organ misplacement" may be more appropriate.

--- We have replaced "organ inversions" with "organ misplacement" as suggested (L. 49 on P. 3).

Q2-6. The paragraph "As CSF is rich in neuropeptides, its orderly flow is critical for nourishing the central nervous system and maintaining proper body axis. Dyskinesia of ependymal multicilia leads to the obstruction of CSF flow, resulting in hydrocephalus and idiopathic scoliosis³⁻⁵." needs attention. Multi in multicilia is unnecessary. Cilia is a plural term. The two sentences, while likely correct individually to some degrees, appear disjoint, illogical together. The first is about ependymal cilia in distributing chemical nutrients, whereas the second talks about ciliary defects will result in physical blockage, presumably increasing ventricular pressure and thus hydrocephaly. Finally, the statement may not be entirely applicable to humans. While hydrocephaly is common in rodents and canines with ependymal cilia dyskinesia, the impact of dyskinesia cilia is often not evident or definitive in humans with such congenital anomalies.

A2-6: Thanks for the valuable comments. We have now replaced "multicilia" by "cilia" throughout the revised manuscript. We have also modified the sentences according to the comments, to read: "As CSF is rich in neuropeptides, its orderly flow is critical for nourishing the central nervous system and maintaining proper body axis. Moreover, it has been reported that improper function of ependymal cilia can lead to the obstruction of CSF flow, resulting in hydrocephalus in rodents, canines and zebrafish³⁻⁵. Recently, dysfunctions in ependymal motile cilia have been shown to lead to phenotypes resembling idiopathic scoliosis in zebrafish, suggesting a critical role of cilia-driven CSF flow in spine development⁶⁻⁷." (L. 52-57 on P. 3).

Results:

P.4

Q2-7. The definitions for RS head in Zheng et al., 2021 and RS head, head-neck complex, RS monomer and RS dimer are imprecise. In the previous study, the RS head contains only Rsph1, 4a, 9 and 3b. In the current study, 4 more components not entirely restricted to the neck were included. As such the text and the proposed stepwise assembly model includes part of the new components in the RS head. The inconsistency is confusing but can be resolved by simply referring the reconstituted particle in the previous study is a partial RS head, whereas RS monomer should be a head-neck monomer. Likewise, RS dimer should be a head-neck dimer. There are additional components in the RS, regardless of monomer and dimer.

A2-7: We apologize for the confusion. We now refer to the reconstituted complex in the previous study as "RS head core complex", for consistency with our previous paper³. We

have also followed the requests of our reviewer and referred to the current two reconstituted structures as “RS head-neck monomer” and “RS head-neck dimer” throughout the revised manuscript.

Q2-8. Figure 1 can be presented in a more persuasive way, to justify the corresponding text and its legend - Cryo-EM structures of mouse RS head-neck complex. Firstly, the legend should state this is reconstituted, ie not naturally isolated, complex. The omission could lead readers to presume they are bona fide RS particles harvested from ciliated cells. Secondly, contrary to the legend, the figure only shows anatomic models, while a cryo-EM montage is relegated to the supplemental data (S1 and S2). Both data sets, preferentially including the overlaid atomic models and representative cryo-EM, are important for readers to assess the claimed 3-4 Å resolution and the fit independently. Furthermore, the recent studies of protist RS^{4,7} (Gui et al., 2021; Grossman-Haham et al., 2021) showed more polypeptides in this region than previously recognized. The original data will allow readers to determine how similar the cryo-EM of the reconstituted particles of only recognized RS proteins resemble the published cryo-EM of RS from various organisms and organs and whether mammalian RS might also have additional components. Modeling should not replace original data.

A2-8: The points have been well taken. We have added “reconstituted” in the revised legend of Fig. 1. In addition, we have moved the model-map fitting for the RS head-neck monomer and dimer from Fig. S3e to Fig. 1 (now as part of Fig. 1c and f). We have also shown the overlaid atomic models and representative cryo-EM densities to illustrate features for both structures (now as Fig. 1e and h). These changes allow readers to assess the 3-to-4 Å resolution and the fit independently.

Regarding the original data of the reconstituted particles, we have presented the original representative micrograph (Supplementary Fig. 1c) and the reference-free 2D class averages (now as Fig. 1b). These original data demonstrate the presence of both the monomer and dimer forms in the reconstituted RS head-neck complex and imply high-resolution structural features, especially in the 2D class averages (Fig. 1b). These data also enable readers to determine how similar the cryo-EM of the reconstituted particles of only recognized RS proteins resemble the published cryo-EM of RS from various organisms and organs.

Furthermore, multi-scale structural comparison between the reconstituted RS head-neck dimer and our *in situ* ependymal cilia RS1/RS2 in the corresponding head-neck region (Fig. 3b-c) suggested that they overall match each other very well especially in the head region, while the lower portion of the neck potentially contains additional component. To make this point clear, we have updated the related description in the revised text, to read “We then fitted our high-resolution cryo-EM map of the reconstituted RS head-neck dimer into the cryo-ET map of RS2 in the head-neck region, and found that they overall match each other very well especially in the head region, while the lower portion of the neck may potentially contain additional component.” (L. 214-217 on P. 8).

Q2-9. The claimed resolution in p. 4 “Our cryo-EM analysis on the assembled mouse RS head-neck complex revealed the presence of both monomer and dimer (Fig. S1C-D). We determined their structures at a resolution of 3.28 Å and 3.57 Å, respectively, with the head core region focus-refined to a resolution of 3.14 Å (Fig. 1B-C, S2, S3A-C, Table S1).” seems inconsistent with Fig S1C-D, which is consistent with the sentence in p. 6 “Our local resolution analysis of the RS head-neck dimer revealed relatively lower local resolution at the very end of the Rsph1 arm and the neck (Fig. S3B)”. Are the stated resolution in both pages inconsistent? Are the stated resolution applicable to Fig. S1C-D?

A2-9: We have checked the resolution evaluation by the 0.143 FSC criterion (Supplementary Fig. 3a), and also regenerated the local resolution map to make the resolution variation easier to be visualized (Supplementary Fig. 3b, also show it below). We confirm that the stated consensus resolution values are all correct and consistent. We have revised the related statement to make it clear, to read: “We then determined the cryo-EM maps of the RS head-neck monomer and dimer at the consensus resolution of 3.28 Å and 3.57 Å, respectively (Supplementary Fig. 3a and Supplementary Table 1). We subsequently focus-refined the individual portions of the two maps, with the head core region of the monomer reached 3.14 Å resolution, and the central bridge as well as the head regions of the dimer reached 3.36/3.46 Å resolution, respectively (Supplementary Figs. 2 and 3a-b, and Supplementary Table 1). Still, the relatively flexible neck regions are at lower resolution (discussed later).” (L. 108-115 on P. 4-5). In addition, the reported resolution was also supported by the representative high-resolution structural features of the two maps (Fig.1e and 1h).

Our local resolution evaluation indicated that most portions of the two maps have higher local resolution than the reported consensus resolution, except for the distal end of the Rsph1 arms and the peripheral portion of the neck, which display relatively lower local resolution (Supplementary Fig. 3b). These regions also appear “fuzzy” in the reference-free 2D class averages (Fig. 1b). Together, these data indicate plasticity in these less constraint regions. We have also incorporated this description in the revised manuscript (L. 192-196 on P. 7).

Supplementary Fig. 3b Local resolution evaluation for the monomer (left), core region of the monomer (middle), and dimer (right) maps.

P. 5

Q2-10. Abbreviations, like XL-MS, should be spelled out when referred the first time. The crosslinker and the properties, at least including the nature of crosslinking and the arm length, should be included for readers to determine independently the validity of proposed molecular proximity and whether crosslinking data “supports” the modeling and the fit as narrated.

A2-10: The point has been well taken and we have added the related information in the revised manuscript, to read “We subsequently conducted cross-linking and mass spectrometry (XL-MS) analysis on the RS head-neck complex by using the amine-to-amine crosslinker BS3 (bis (sulfosuccinimidyl) suberate), which has a spacer-arm-length of 11.4 Å⁴³⁻⁴⁴” (L. 133-135 on P. 5).

Q2-11. In “wrapped around by the previously missing N-terminal Dpy-30 motif of the head subunits Rsph4a and Rsph4a’ (Fig. 1D)”, the underline space may benefit from the word “degenerate” and the reference. The DPY-30 motif in RSP4 was revealed by structural studies (Gui et al., 2021) and cannot be found by mere protein sequence analysis. Fig. 1D is not helpful in this regard.

A2-11: Thanks for the suggestion. We have added the reference to the *Chlamydomonas* head-neck structure in the model building part, as follows: “We then built atomic models for the RS head-neck monomer and dimer (Fig. 1d, g) based on the mouse RS head core structure²⁷ and AlphaFold2^{41,42} predicted models for the neck subunits mostly based on the *Chlamydomonas* RS head-neck structures^{28,29}, with constraints of the current maps.” (L. 117-120 on P.5). Moreover, to avoid the redundancy in the detailed subunit interaction networks, we simplified the description of the Rsph3d interaction network by removing the motif-level description, to read: “Rsph3b is attached to one side of the head and is intimately associated and stabilized by the neck subunits Rsph2 and Rsph23, as well as the head subunits Rsph4a and Rsph4a’ (Fig. 1d).” (L. 126-128 on P. 5).

Q2-12. “Unlike the extensive interplays among the other subunits, however, Rsph10b only exhibits limited association with Rsph16 in the XL-MS results (Fig. 1F), suggesting that Rsph10b may not be a standard component of RS1 and RS2’s head-neck complex.” This sentence is confusing. It assumes that extensive interplays and/or extensive association with Rsph16 is a feature of core head-neck proteins. Also, core may be a better word than standard.

A2-12: We apologize for the confusion. Our intention was to imply that Rsph10b is not a component of RS1 and RS2’s head-neck complex, thus foreshadowing our subsequent assignment of Rsph10b to the head of RS3 (Fig. 3f). To improve the clarity, we have modified the sentences in the revised manuscript as “Notably, although Rsph10b appeared to be moderately abundant in the affinity-purified reconstituted RS head-neck complex (Supplementary Fig. 1b), it was only weakly detected by XL-MS (Fig. 1i) and completely absent in the RS1/RS2 head-neck maps (Fig. 1c, f) after subjecting the sample to an additional round of glycerol gradient centrifugation. Such results suggest that Rsph10b and

some of the head-neck subunits may form a complex distinct from the head-neck complex of RS1/RS2.” (L. 137-142 on P. 5-6).

Q2-13. The narrations before and after the section title “Subunit interaction network of the RS head-neck complex” in p. 5, seem to contain redundant descriptions about the key proteins and referral to the preceding publication. Reorganization may lessen the redundancy. In addition, it is unclear whether most descriptions in this section about conserved core proteins are any different from that in algal counterpart (Gui et al., 2021). It is good to have a summary statement and highlight the differences now, rather than later.

A2-13: We thank our reviewer for the comments. Following the reviewer's suggestions, we have re-organized the related main text and edited our descriptions to increase the clarity and readability of the manuscript. In addition, we have added a new paragraph to summarize and discuss the differences between the conserved core proteins of mammalian and *Chlamydomonas*^{4,7} in the revised manuscript (L. 159-169 on P. 6).

P. 7

Q2-14. The conclusion of the middle paragraph, “Collectively, these data suggested that our recombinant RS head-neck complex represents the endogenous head-neck architecture of both RS1 and RS2, thus further confirming that the mammalian RS head complex is distinct in composition from the *Chlamydomonas* one^{7,23-25,27}.” is perplexing. The underlined phrase conflicts with the established facts and previous statements naming human spoke components after algal counterparts founded on extensive homologies in structural motifs and sequence similarities among orthologs throughout evolution. Wording with precision in this case is necessary to reflect the facts and does not diminish the contribution of this study.

A2-14: In response to the comments, we have revised the statement as follows: “Collectively, these data suggest that our recombinant RS head-neck complex represents the endogenous head-neck architecture of both RS1 and RS2, thus further confirming that the mammalian RS1/RS2 head-neck is simplified in composition compared to its counterpart in *Chlamydomonas*^{8,27-29,32}.” (L. 218-221 on P. 8).

P. 8

Q2-15. In “...we then fitted Cfap61 and Cfap251, the homolog of Fap61 and Fap251”, it is not clear at all if C is indicative of *Tetrahymena*, *Chlamydomonas*, Human or Mice. The paragraph in current state does not make sense.

A2-15: We are sorry for causing the confusion. In literature, FAP is an abbreviation for Flagella-associated protein, whereas CFAP or Cfap is for Cilia- and flagella-associated protein. For historic reasons, *Chlamydomonas* cilia are preferably called flagella, and thus FAP has been used as a prefix to name some of their proteins. Accordingly, for other species containing cilia, CFAP or Cfap is commonly used as the prefix when referring to a FAP counterpart. This nomenclature, however, is not followed by everyone. This is mainly why our descriptions lead to the confusion because we tried to stick to the names used in

each cited paper. In the revised manuscript, we have modified the text and used C_{fp} as the prefix together with species names to avoid confusion (L. 268-274 on P. 10).

P. 9

Q2-16. In “two RS1 conformations relative to the DMT”, conformation, meaning form or shape, is inappropriate for the slight tilt of the complex.

A2-16: Thanks for pointing this out. We have changed the “conformation” to “orientation”. (L. 284 on P. 10)

Q2-17. In “... RS1 may be more dynamic in regulating cilia motion...”, it is unclear whether the authors imply RS1 is inherently more dynamic like changing position or conformation, or merely structurally more flexible, or is more susceptible to induced tilt upon interacting with the central pair apparatus, as shown in another ciliated organism (Warner and Satir, 1974).

A2-17: We appreciate the insightful comments from our reviewer. We aim to convey that RS1 is inherently more dynamic, exhibiting larger conformational space with RS1 in perpendicular, tilted, and occasionally downward tilted orientations relative to the attached DMTs. We have rephrased this sentence accordingly, to read “Taken together, RS1 might be more dynamic, exhibiting larger conformational space with RS1 in perpendicular, tilted, and occasionally downward tilted orientations, likely due to the lack of constraint from its neighboring RSs. In contrast, RS2 and RS3 are mostly perpendicular and relatively stable, which could be attributed to the mutual contact in their head region in mammalian cilia^{16,18}. We then postulate that RS1 may be more dynamic in regulating ependymal cilia motion.” (L.282-287 on P. 10).

Q2-18. “In contrast to the helical beat form of sperm flagella, mammalian ependymal and respiratory multicilia beat in a back-and-forth manner but differ greatly in number.....”. This sentence is so simplified that it becomes inaccurate. Sperm flagella exhibit a variety of waveform, depending on the species, the activation state, and the environments, for example. The term helical beat form, which implies 3-D, does not do the justice. If ciliary motility parameters are not the focus of this study, it suffices to state simply differences in waveform, number.....

A2-18: Following the request, we have revised our statement as follows: “Although both ependymal cilia and tracheal cilia exhibit a coordinated metachronal waveform, they differ in their beat frequencies. Ependymal cilia beat at a frequency of around 34 Hz at 37°C, whereas tracheal cilia at approximately 19 Hz⁵⁸. Sperm motility arises from the beating of a single flagellum capable of a variety of waveforms, depending on species, activation state, and environments, at a frequency of up to 20 Hz⁵⁹.” (L. 289-293 on P.10).

P. 10

Q2-19. “...the mouse sperm flagella exhibit much more filamentous MIPs that is similar to but more extensive than that of tektin filaments inside the A-tubule of DMT (Fig. 4C)22.” are?

A2-19: Sorry for the typo. We also realized that the sentence lacks clarity and completeness. In the revised manuscript, we have modified the description as follows: “In contrast to the mouse ependymal cilia, mouse sperm flagella exhibit many more filamentous MIPs, such as the Tektin bundle and other sperm-specific filaments inside the A-tubule and striated densities attached to the inner wall of the B-tubule²⁵ (Fig. 4c).” (L. 318-321 on P.11).

Q2-20. In the N-DRC paragraph, it is unclear what the main point from the exhaustive structural descriptions is. What do they mean? Is it same as or different from that in other systems and organisms and systems? The detail structures of the axoneme cannot fit into one paper. This section could go to supplemental data, if it does not add to the coherent story.

A2-20: We appreciate the comments. We compared our *in situ* N-DRC map of ependymal cilia with the *in situ* N-DRC maps of *Chlamydomonas* flagella (EMD-20338)⁵ and human respiratory cilia (EMD-5950)⁶, respectively. Unlike the other two maps, the ependymal cilia N-DRC exhibits a reduced distal lobe density containing the C-terminal portions of DRC9/10 (please refer to Fig. R3 on page 7 of this responding letter). This suggests a weakened interaction between DRC9/10 and the neighboring DMT in ependymal cilia.

Following the suggestion, we have moved the contents in Fig. 4D into supplemental data (now as Supplementary Fig. 8d in the revised manuscript). In the figure caption, we briefly mention that “the distal right lobe density is rather weak” (L. 895-898 on P. 45).

P. 11

Q2-21. In “.... collectively indicating a potentially more complex RS-CP interaction mode in mammals” it is not evident from the sentence why the authors think the different chemical forces in RS-CO interactions means more complex in mammalian modes.

A2-21: Thanks for the comment. We have removed this statement.

Discussion

Q2-22. It seems that in p. 11 “...intrinsic modes of motion of this regulatory machinery of ciliary motility (Fig. 2A, C).” and “the intrinsic dynamics of the head-neck dimer” in the Fig. 2 legend convey the same concept. Intrinsic dynamics seems to be different from intrinsic flexibility and imply inherent mobility. Albeit possible, this cannot be demonstrated by solid phase imaging in this study. Additional modest interpretations should be included, like the structural variations imply flexibility which may allow RS transduce forces between the central pair and DMTs as cilia in natural environments bend or twist, not limited to rhythmic beating with identical waveform and beat frequency.

A2-22: In the current study, we used the 3D Flexible Refinement (3DFlex) in cryoSPARC⁸, a generative neural network method, to determine the continuous motion of the RS head-neck dimer from cryo-EM images. Regarding whether cryo-EM images can reveal intrinsic dynamics, it has been stated in this method paper⁸ that: “Single-particle cryo-EM collects thousands of static 2D particle images that, in aggregate, may span the target protein’s 3D

conformational space”, and “From two-dimensional image data, 3DFlex enables the determination of high-resolution 3D density, and provides an explicit model of a flexible protein’s motion over its conformational landscape”. Therefore, they pointed out that “Cryo-EM holds great promise for uncovering both the high-resolution structure and motion of biologically functional moving parts”. This is why we used the terms of “intrinsic modes of motion” and “intrinsic dynamics” in our description. Still, we appreciate the constructive suggestion from our reviewer and have replaced “intrinsic dynamics” to “intrinsic flexibility” throughout the manuscript. We have also added the suggested modest interpretations in the end of the section “Conformational flexibility of the RS head-neck dimer” (L. 203-206 on P. 8).

Q2-23. In “... sperm-specific component densities cross-linking RS1/RS2/RS3 were absent...”, alternative terms - like connecting or linking - may be more suitable than cross-linking, given cross-linking is used in XL-MS.

A2-23: We have replaced “cross-linking” with “connecting” in the revised manuscript (L. 352 on P. 12).

Q2-24. “Collectively, our study sheds new lights on the stepwise assembly mechanism of mammalian RS1/RS2...” overstated. The authors certain earn the right to propose a stepwise assembly model. Experiments are needed in this case to shed the light.

A2-24: The point has been well taken. We have rephrased the description, to read “We also identified an extra subunit..., and proposed a stepwise assembly mechanism of mammalian RS1/RS2...” (L. 358-359 on P. 12).

P. 12

Q2-25. “Noteworthy, Rsph16, the homologue of HSP40 and also comprising a C-terminal dimerization domain³⁵, appears to play an essential role in the dimerization of the RS head-neck complex, which is key for the formation and stabilization of the RS1/RS2.” The structural evidence in this study did not provide evidence in assembly, ie the formation. In fact, the interpreted essential role in the formation is not supported by the genetic evidence. Contrary to the prediction, RS1 and RS2 in algal mutants of RSP16 display no evident defects in compositions of RS1 and RS2, other than sporadic aberrant morphology and tilts.

A2-25: We appreciate this valuable comment. We have deleted “which is key for the formation and stabilization of the RS1/RS2” in the revised manuscript. As the last step in the paragraph (“(6) then the RS head-neck dimer can bind to the base/docking component...”) is beyond the proposed assembly mechanism for the RS head-neck complex, we have also re-organized the paragraph to improve the clarify of our presentation. (L. 363-381 on P. 13)

P. 13

Q2-26. “... ependymal cilia may not require tektins within the A-tubule to provide extra stabilization/stiffness of DMT against waveform-dependent mechanical forces.” is awkward.

The preceding statements are about less viscous environment, not about waveform. A revision is needed to make the argument sensible.

A2-26: The point has been well taken. We have modified the description accordingly, to read: "... to handle the less viscous environment of CSF, ependymal cilia may not require Tektins within the A-tubule of DMT to provide extra stabilization/stiffness of DMT" (L. 411-413 o P. 14).

Q2-27. In "the IDAs which generate the waveform of..." perhaps determine is more appropriate.

A2-27: We have replaced "generate" with "determine" in our revised manuscript (L. 416 on P. 14).

Q2-28. For "Although it has been reported that IDA-b/e are missing from at least two DMTs in mouse respiratory cilia⁶⁵, and one of the DMTs bears lost IDA-c in Chlamydomonas, the complete loss of IDA-b/c/e in all DMTs has not been observed in any other tissues.", the underlined statement needs a reference and revision. "bears" does not make sense.

A2-28: We have added the related reference and replaced "bears lost" with "lacks". We noticed that the "IDA-c" should be "IDA-b" and have corrected the mistake in the revised manuscript (L. 421 on P. 14).

P. 14

Q2-29. The middle paragraph is illogical in the current state. The authors seem to use the different amounts of MIPs and IDA species to explain the differences in the environment and required function of mammalian sperm flagella that need to navigate tortuous female reproductive track, cilia in viscous environments and ependymal cilia in CSF of low viscosity. But refinements are needed to make the argument understandable.

A2-29: Following the request, we have modified this paragraph as follows: "It is also interesting to note that, from mammalian sperm flagella to respiratory cilia then to ependymal cilia, ciliary composition and ultrastructure become increasingly simplified (Fig. 4a-c and Supplementary Fig. 8a-c)^{18,25}. These simplifications might be related to the differences in their working environments and functions. After all, mammalian sperm flagella need to navigate through the intricate female reproductive tract⁷², respiratory cilia beat in viscous mucous, and ependymal cilia beat in watery CSF. Detailed structure-function relationships, however, remain to be clarified in the future." (L.431-437 on P.15).

A2-30. The authors should discuss the possibility that the missing key structures in cilia of cultured ependymal cells may only happen in culture.

A2-30. We appreciate this insightful comment. Since cilia in wild-type airway epithelial cells that are induced to differentiate into multiciliated cells in culture have not been found to show intrinsic ultrastructural defects⁶, it is unlikely that the cilia in cultured wild-type EPCs would be born to be defective. Furthermore, no publication to date has implicated a structural difference between cilia from cultured EPCs and ependymal tissues. Nevertheless, as stated in our responses to reviewer #1's comments (A1-1 and A1-2), we

agree that we cannot absolutely exclude the possibility that the missing key structures are due to the sample preparation, we have included a statement in the Discussion section to indicate this possibility (“Alternatively, the absence of certain key structures might result from cell separation or cryo-vitrification due to the difficulty to absolutely exclude this possibility at present”) (L.428-430 on P.14-15).

Materials and Methods

P. 18

Q2-31. Reference is needed for “Mouse EPCs were isolated and cultured as described.”

A2-31: We have now added the requested reference⁹ after this statement.

References

- 1 Walton, T. *et al.* Axonemal structures reveal mechanoregulatory and disease mechanisms. *Nature* **618**, 625-633 (2023). <https://doi.org:10.1038/s41586-023-06140-2>
- 2 Ryan, R. *et al.* Functional characterization of tektin-1 in motile cilia and evidence for TEKT1 as a new candidate gene for motile ciliopathies. *Human Molecular Genetics* **27**, 266-282 (2018). <https://doi.org:10.1093/hmg/ddx396>
- 3 Zheng, W. *et al.* Distinct architecture and composition of mouse axonemal radial spoke head revealed by cryo-EM. *Proceedings of the National Academy of Sciences* **118** (2021). <https://doi.org:10.1073/pnas.2021180118>
- 4 Gui, M. *et al.* Structures of radial spokes and associated complexes important for ciliary motility. *Nature Structural & Molecular Biology* **28**, 29-37 (2020). <https://doi.org:10.1038/s41594-020-00530-0>
- 5 Gui, L. *et al.* Scaffold subunits support associated subunit assembly in the Chlamydomonas ciliary nexin-dynein regulatory complex. *Proc Natl Acad Sci U S A* **116**, 23152-23162 (2019). <https://doi.org:10.1073/pnas.1910960116>
- 6 Lin, J. *et al.* Cryo-electron tomography reveals ciliary defects underlying human RSPH1 primary ciliary dyskinesia. *Nature Communications* **5**, 5727 (2014). <https://doi.org:10.1038/ncomms6727>
- 7 Grossman-Haham, I. *et al.* Structure of the radial spoke head and insights into its role in mechanoregulation of ciliary beating. *Nat Struct Mol Biol* **28**, 20-28 (2021). <https://doi.org:10.1038/s41594-020-00519-9>
- 8 Punjani, A. & Fleet, D. J. 3DFlex: determining structure and motion of flexible proteins from cryo-EM. *Nature Methods* **20**, 860-870 (2023). <https://doi.org:10.1038/s41592-023-01853-8>
- 9 Liu, H. *et al.* Wdr47, Camsaps, and Katanin cooperate to generate ciliary central microtubules. *Nat Commun* **12**, 5796 (2021). <https://doi.org:10.1038/s41467-021-26058-5>

REVIEWERS' COMMENTS

Reviewer #1 (Remarks to the Author):

The authors responded all the points from this reviewer sincerely. Their additional experiments were very convincing and clarified questions. It seems they addressed the other reviewer's point, too. This reviewer strongly recommend publication of this manuscript in Nature Communications.

Reviewer #2 (Remarks to the Author):

The revised manuscript entitled “Multi-scale structures of the mammalian radial spoke and divergence of axonemal complexes in ependymal cilia” is strengthened significantly in the Results and writing. The current version is suitable for the publication in the Nature Communication and will be of interest to researchers and clinicians alike.

An optional suggestion is to change “Strikingly” in the Abstract into “Contrary to the conserved core radial spoke structure, ...”. It may not be evident to readers what revealed by cryotomography are striking, whereas the latter lacking any dramatic wording may be more illuminating and bridging two seemingly separate projects into a coherent paper.

REVIEWERS' COMMENTS

Reviewer #1 (Remarks to the Author):

Comments: The authors responded all the points from this reviewer sincerely. Their additional experiments were very convincing and clarified questions. It seems they addressed the other reviewer's point, too. This reviewer strongly recommend publication of this manuscript in Nature Communications.

Responses: We wish to express our appreciation for the time and effort our reviewer dedicated to evaluating our work. The insightful comments and valuable feedback provided has significantly contributed to improving the quality of our research.

Reviewer #2 (Remarks to the Author):

Comments: The revised manuscript entitled “Multi-scale structures of the mammalian radial spoke and divergence of axonemal complexes in ependymal cilia” is strengthened significantly in the Results and writing. The current version is suitable for the publication in the Nature Communication and will be of interest to researchers and clinicians alike.

Responses: We sincerely appreciate our reviewer for the thoughtful and positive evaluation of our revised manuscript. The professional and constructive comments, along with valuable advice throughout the review process, have significantly contributed to the improvement of our work, enabling us to present it more effectively.

Comments: An optional suggestion is to change “Strikingly” in the Abstract into “Contrary to the conserved core radial spoke structure,”. It may not be evident to readers what revealed by cryotomography are striking, whereas the latter lacking any dramatic wording may be more illuminating and bridging two seemingly separate projects into a coherent paper.

Responses: Thanks for this constructive suggestion. We have replaced 'Strikingly' with 'Contrary to the conserved radial spoke structure' in the Abstract of the manuscript.